# Recursive Self-Aggregation Unlocks Deep Thinking in Large Language Models

## Abstract

Test-time scaling methods improve the capabilities of Large Language Models (LLMs) by increasing the amount of compute used during inference to make a prediction. Inference-time compute can be scaled *in parallel* by choosing among multiple independent solutions or *sequentially* through self-refinement. We propose Recursive Self-Aggregation (RSA), a test-time scaling method inspired by evolutionary methods that combines the benefits of both parallel and sequential scaling. Each step of RSA refines a population of candidate reasoning chains through aggregation of subsets to yield a population of improved solutions, which are then used as the candidate pool for the next iteration. RSA exploits the rich information embedded in the reasoning chains – not just the final answers – and enables bootstrapping from partially correct intermediate steps within different chains of thought. Empirically, RSA delivers substantial performance gains with increasing compute budgets across diverse tasks, model families and sizes. Notably, RSA enables Qwen3-4B-Instruct-2507 to achieve competitive performance with larger reasoning models, including DeepSeek-R1 and o3-mini (high), while outperforming purely parallel and sequential scaling strategies across AIME-25, HMMT-25, Reasoning Gym, LiveCodeBench-v6, and SuperGPQA. We further demonstrate that training the model to combine solutions via a novel aggregation-aware reinforcement learning approach yields significant performance gains. Code is available here.

## 1 Introduction

Large language models (LLMs) demonstrate consistent improvements in performance with increasing training compute (Kaplan et al., 2020). Complementarily, *test-time scaling* strategies, *i.e.*, those that increase compute at inference without altering model parameters, can deliver significant gains in performance (Snell et al., 2025; Jaech et al., 2024). Test-time scaling mechanisms for LLMs can broadly be characterised into two types (§2): those that use deeper model rollouts to iteratively improve solutions (*e.g.*, Muennighoff et al., 2025; Zhang et al., 2025a) and those that branch to explore multiple solution paths, then filter or recombine them (*e.g.*, Wang et al., 2023; Weng et al., 2023). We refer to these types as *sequential* and *parallel* scaling; some *hybrid* methods combine the strengths of both approaches (*e.g.*, Yao et al., 2023; Meyerson et al., 2024; Lee et al., 2025).

However, a universal and effective test-time-scaling method that allows reuse of promising fragments from multiple candidate solutions is lacking. Self-refinement methods – the quintessential form of sequential scaling – can improve a candidate solution by reusing its own correct parts, but do not leverage the information contained within other candidates. Similarly, parallel scaling methods such as verifier-guided Best-of-N selection can identify the best candidate from a batch, but do not recombine candidates to produce improved solutions. Existing hybrid approaches fail to solve this problem in a general way, often making strong assumptions on the form of reasoning chains (*e.g.*, Meyerson et al., 2024; Hemberg et al., 2024) or requiring external verifiers (*e.g.*, Novikov et al., 2025; Lee et al., 2025). Our work fills this gap in three ways, described in the following paragraphs.

**Self-aggregation.** We study a general way to improve LLM reasoning chains through *self-aggregation*: providing the model with the query and a set of candidate solutions and prompting it to produce an improved solution. Such an approach, which relies on the implicit verification abilities of the model (Weng et al., 2023), can use the rich information contained within the reasoning chains: for example, a reasoning trace that results in an incorrect answer to a problem can have correct

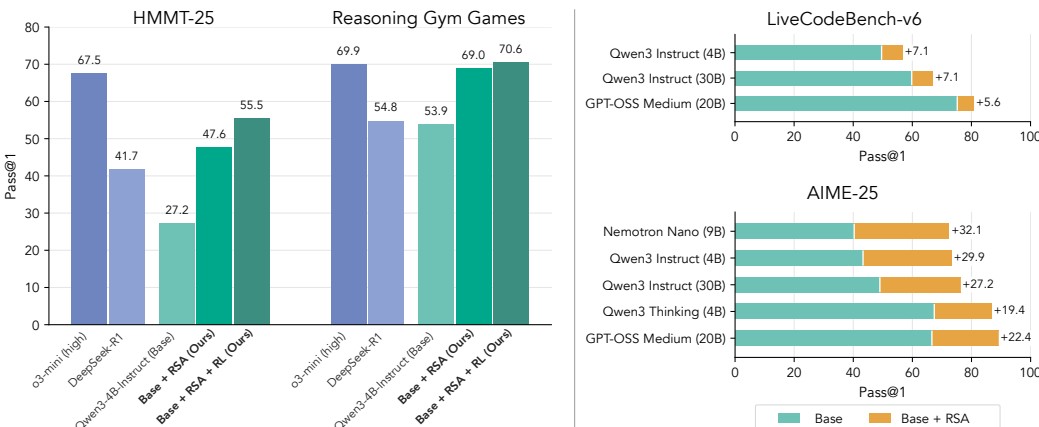

Figure 1: **Recursive Self-Aggregation (RSA, §3) substantially improves Pass@1 across tasks and model architectures.** RSA enables the much smaller Qwen3-4B-Instruct-2507 to match the performance of larger reasoning models such as DeepSeek-R1 and o3-mini (high). These gains are further amplified through our proposed aggregation-aware RL framework (§4).

intermediate steps that can be reused in the aggregated solution (§F). Such aggregation methods are explored in multiple concurrent works (*e.g.*, Li et al., 2025; Wang et al., 2025; Zhao et al., 2025) and are promising directions for test-time scaling.

**Recursive self-aggregation.** While self-aggregation can be used as a one-time procedure to combine candidate solutions, our proposed algorithm, *Recursive Self-Aggregation* (RSA, §3), goes further: integrating aggregation steps into a self-improvement loop motivated by evolutionary algorithms. RSA maintains a population of candidate solutions and iteratively recombines subsets of the population to produce a new population of improved solutions (Fig. 3). This sequential refinement enables deeper reasoning by allowing the model to revisit its solutions and make multiple attempts at correcting errors. RSA maintains a candidate population larger than the aggregation set size and can therefore jointly consider significantly more proposals than single-step aggregation, which is constrained by the model's effective context length. Unlike other evolutionary methods, RSA requires no external verification and can be seamlessly integrated into any LLM inference pipeline to improve reasoning.

**Aggregation-aware RL.** During post-training, LLMs are trained with reinforcement learning (RL) to improve their reasoning ability (Jaech et al., 2024; Guo et al., 2025). RL training does not account for test-time scaling, which in this case is the task of aggregating multiple reasoning chains. In fact, we observe that standard RL fine-tuning can even degrade performance relative to the base model when combined with test-time aggregation (§5.4). To address this, we propose an *aggregation-aware* RL approach using a simple data-augmentation strategy to train LLMs to aggregate solutions (§4).

We perform extensive experiments to demonstrate the effectiveness of RSA across diverse tasks, such as AIME-25, HMMT-25, LiveCodeBench, Reasoning Gym, and SuperGPQA with various base models (§5). RSA bridges the gap between the lightweight Qwen3-4B-Instruct-2507 and much larger reasoning models like DeepSeek-R1 and o3-mini (high) (Fig. 1). Our results also show that aggregation-aware RL significantly improves performance with RSA compared to naïve RL training (§5.4). We rigorously analyze the factors driving RSA performance and provide practical recommendations to enable deeper test-time thinking under compute constraints (§5.3).

## 2 A TAXONOMY OF TEST-TIME SCALING METHODS

*Test-time scaling* refers to methods that obtain predictions using a static LLM with a larger number of model evaluations than that required by simply prompting for an answer. These methods significantly improve performance without modifying model weights (Snell et al., 2025; Zhang et al., 2025b), effectively using the model as a component in an external optimization or inference framework, at the cost of increased computation.

A well-designed test-time scaling framework should yield monotonic improvements in performance as compute budgets increase, similar to scaling laws for pretraining (Kaplan et al., 2020; Snell et al., 2025). Most methods rely on some kind of verification, whether implicit or explicit, incorporated within a sequential or parallel control flow. In this section, we review the literature on test-time scaling

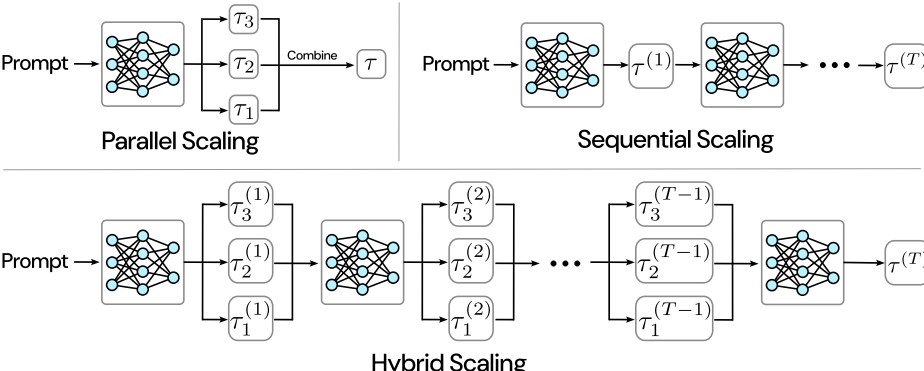

Figure 2: **Overview of test-time scaling control flows.** *Parallel* methods generate multiple candidates and select using a verification mechanism. *Sequential* methods iteratively refines a chain, correcting previous mistakes. *Hybrid* methods combine parallel branching with sequential refinement.

in LLMs and provide a taxonomy of test-time scaling frameworks based on the verification strategy and control flow they employ, illustrated in Fig. 2. Building on this, we then introduce our proposed approach, *Recursive Self-Aggregation* (RSA) in §3. See §B for a discussion of broader related work.

## 2.1 VERIFICATION STRATEGY

**External verification.**   Any external optimization procedure requires a mechanism to assess the quality of proposed solutions. In domains such as code or math, evaluation can often be performed exactly using external tools (*e.g.*, compilation and execution (Gao et al., 2023b)). When such verifiers are unavailable, inference-time feedback is instead obtained via *learned* reward models, trained on preference data or correctness signals derived from reasoning chains (Cobbe et al., 2021; Ouyang et al., 2022; Snell et al., 2025). This verifier feedback, exact or learned, makes it possible to improve solution quality as more compute is allocated: a simple strategy is Best-of-N (Gao et al., 2023a), where $N$ candidates are generated and the highest-reward solution is selected.

**Self-verification.**   LLMs exhibit a generation-verification gap: they are more reliable at judging correctness of solutions than producing them (Li et al., 2024). This property can be exploited to enable test-time scaling by using the LLM as a verifier of its own outputs (*e.g.*, Madaan et al., 2023; Weng et al., 2023). The LLM can also be further fine-tuned to enhance its verification ability (Zhang et al., 2025a), but we regard this as a form of external verification since it requires learning a verifier.

**Implicit verification.**   Some methods bypass explicit verification by relying on the LLM to generate improved solutions, effectively performing verification of solutions without scoring them. For example, majority voting (Wang et al., 2023) works on the assumption of self-consistency: that the model produces correct answers more consistently than incorrect ones. Similarly, frameworks for self-refinement (*e.g.*, Madaan et al., 2023) iteratively refine a reasoning chain without being explicitly prompted for verification. RSA falls within this category: rather than explicitly verifying each solution, the model implicitly checks intermediate steps across multiple reasoning chains, allowing it to correct errors and generate improved solutions.

## 2.2 REASONING CONTROL FLOW

**Parallel scaling.**   These strategies generate multiple independent reasoning chains in parallel and then combine them to yield the final answer. Typical procedures for combination include majority voting, Best-of-N selection or single-step aggregation of the sampled proposals (Wang et al., 2023; Snell et al., 2025; Li et al., 2025). These strategies rely on the inherent diversity in sampling from the LLM, allowing parallel proposals to efficiently explore the search space and allow optimal GPU memory utilization. These algorithms embody the philosophy of *breadth-first thinking*.

**Sequential scaling.**   Purely parallel scaling sacrifices the ability to think deeply, which is often crucial for multi-step reasoning tasks that cannot be solved efficiently through guess-and-check. Sequential scaling instead increases the number of iterative model evaluations to produce higher-quality solutions, for example, by inducing a model to correct errors in its reasoning (Muennighoff et al., 2025) or simply increasing the number of latent reasoning tokens it can generate. While these strategies generally require more computation time than parallel ones (given sufficient memory budgets), they are well suited to complex reasoning problems requiring *depth-first thinking*. However,

Figure 3: RSA generates a population of $N$ solutions for a given prompt and recursively updates them over $T$ steps. Each update step subsamples $K$ distinct solutions from the current population and generates an improved solution with the aggregation prompt. See §C for algorithm pseudo-code.

the lack of branching in such methods limits their ability to explore alternative continuations of promising solution paths, making the model prone to persisting in an unproductive reasoning chain. Sequential scaling also leaves excess GPU memory underutilized.

**Hybrid scaling.** Sequential and parallel scaling strategies can be combined in hybrid frameworks that draw on the strengths of both paradigms. These methods make efficient use of GPU memory by evaluating many candidate solutions in parallel, while also incorporating sequential depth to iteratively refine and improve the batch of solutions. One strong class of hybrid methods uses LLMs as components within a genetic algorithm loop (*e.g.*, Novikov et al., 2025; Lee et al., 2025; Meyerson et al., 2024), all using external verification to score candidates. Another example of hybrid test-time scaling is Mixture-of-Agents (Wang et al., 2024), where an ensemble of LLMs generates improved proposals that are aggregated by a strong model into the seed solution for the next iteration. Our method, RSA, is also a hybrid scaling algorithm: like Mixture-of-Agents, it relies on recursive aggregation, but it further maintains a population of candidate solutions larger than the aggregation batch size, similar to evolutionary algorithms, while only using a single LLM. By aggregating random subsets of this population, RSA preserves diversity in the candidate pool, which is critical when all proposals and aggregations are produced by the same model (as studied in §5.3 and §D).

## 3 EVOLVING THOUGHTS USING RECURSIVE SELF-AGGREGATION

We present *Recursive Self-Aggregation* (RSA), a hybrid test-time scaling procedure designed to improve the model's performance without complex scaffolding or using external verifiers. It frames reasoning as a form of evolutionary process where candidate reasoning chains are iteratively refined through self-aggregation, inspired by the crossover and mutation steps in genetic algorithms. RSA is simple to implement and leads to substantial improvements in reasoning abilities across different models and tasks, when compared to pure sequential or parallel scaling (§5). Fig. 3 illustrates the core components of RSA, which we describe below. The algorithm is also written in §C.

Given a query $\mathbf{x}$ and a pretrained LLM $p_{\theta_{\mathrm{ref}}}$, RSA maintains a population of $N$ candidate solutions $\mathcal{P}_t$ at each step $t$. The model is provided with the question and a subset of $K$ solutions from this population, and prompted to produce an improved population of solutions $\mathcal{P}_{t+1}$. The procedure is described in detail below:

1. **Population of trajectories.** At any given step $t$, RSA maintains a population of $N$ independent candidate solutions $\mathcal{P}_t := \{\tau_1^{(t)}, \ldots \tau_N^{(t)}\}$. The initial population $\mathcal{P}_1$ is generated by sampling $N$ responses for query $\mathbf{x}$ using the LLM $p_{\theta_{\mathrm{ref}}}$:

$$\tau_i^{(1)} \sim p_{\theta_{\mathrm{ref}}}(\cdot \mid \mathbf{x}), \quad \mathcal{P}_1 = \{\tau_1^{(1)}, \ldots, \tau_N^{(1)}\}. \tag{1}$$

2. **Subsampling**. We form $N$ aggregation sets of $K$ candidates, where each set is sampled uniformly without replacement from the population:

$$\mathcal{S}_t = \{S_1^{(t)}, S_2^{(t)}, \ldots, S_N^{(t)}\}, \quad S_i^{(t)} \subseteq \mathcal{P}_t, \ |S_i^{(t)}| = K. \tag{2}$$

3. **Aggregation.** Each set $\mathcal{S}_i^{(t)}$ and the query $\mathbf{x}$ is formatted using an aggregation prompt directing the LLM $p_{\theta_{\mathrm{ref}}}$ to generate a refined response $\tau_i^{(t+1)}$, forming a new population of candidates $\mathcal{P}_{t+1}$:

$$\tau_i^{(t+1)} \sim p_{\theta_{\mathrm{ref}}}(\cdot \mid S_i^{(t)}, \mathbf{x}), \quad \mathcal{P}_{t+1} = \{\tau_1^{(t+1)}, \ldots, \tau_N^{(t+1)}\}. \tag{3}$$

RSA recursively updates the population $\mathcal{P}_t$ using (2) and (3) for $t = 1, \ldots, T - 1$. This sequential loop is expected to allow errors and inconsistencies to be gradually pruned away during aggregation, while preserving favorable reasoning patterns. Consequently, we expect overall diversity within the population to generally decrease as $t$ increases, accompanied by a monotonic improvement in success rate (See §D).

4. **Termination.** Given the final population of candidate solutions $\mathcal{P}_T$, the solution is obtained either by randomly sampling from this population or by majority voting. We use uniform random sampling in all our experiments, to evaluate our method without any special selection mechanism.

Note that the intermediate trajectories $\tau_i^{(t)}$ are not required to terminate with complete answers; even partial reasoning chains can provide valuable signal during aggregation. Additionally, the choice of $K$ defines the number of alternative responses to consider for aggregation, with $K = 1$ being equivalent to sequential self-refinement (Madaan et al., 2023). In §5.3, we show that even setting $K = 2$ leads to significant improvements over self-refinement, highlighting the importance of combining diverse solutions for improving reasoning performance. See §F for an illustrative example of aggregation.

An important consideration is that self-aggregation can lead to loss of diversity due to excessive reuse of reasoning patterns that occur in multiple trajectories in the population. Maintaining a large population size $N$ relative to the aggregation size $K$ helps ensure sufficient diversity for recombination. However, a very large $N$ relative to $K$ can lead to slow convergence of the population as a whole, since high-quality reasoning patterns will require more iterations to dominate the population. We study these tradeoffs in §5.3. The aggregation prompts we use are provided in §G.

## 4 TRAINING AGGREGATORS WITH REINFORCEMENT LEARNING

In addition to the test-time strategies discussed thus far, a model's reasoning ability can be improved by post-training it with reinforcement learning (RL) (Jaech et al., 2024; Guo et al., 2025). Standard RL post-training encourages a model to produce correct solutions, conditioned on the query (Trung et al., 2024; Lambert et al., 2025). While this improves the model's ability to directly generate correct solutions, it does not explicitly teach the model how to aggregate multiple candidate solutions. As we show in §5.4, this mismatch between the training objective and the test-time strategy can result in *worse* performance compared to the base (reference) model when using RSA.

To better align training and inference, we formulate the task of aggregation as an RL problem. The reference model $p_{\theta_{\text{ref}}}$ generates a set of candidate reasoning chains given a problem. Next, the model is trained to produce a single correct reasoning chain given the problem and the set of candidate reasoning chains. To achieve this in practice, we create an *aggregation-aware* training dataset consisting of two types of prompts: (1) Standard prompts, containing only the problem, to train the model to propose good initial candidate reasoning chains; and (2) aggregation prompts, which include the problem along with $K$ candidate solutions from the reference model, formatted with the same aggregation prompt used for RSA; see §G.

Consider problem-solution pairs sampled from some dataset $(\mathbf{x}, \mathbf{y}) \sim \mathcal{D}$, and candidate solutions $\tau$ generated by the model conditioned on the problems. Training with the standard prompts described above corresponds to the standard RL training of LLMs that optimizes the following objective:

$$\max_{\theta} \mathbb{E}_{(\mathbf{x},\mathbf{y})\sim\mathcal{D}} \left[ \mathbb{E}_{\tau\sim p_{\theta}(\cdot|\mathbf{x})} \left[ r(\tau, \mathbf{y}) \right] - \beta \, \text{KL} \left( p_{\theta}(\cdot \mid \mathbf{x}) \,\|\, p_{\theta_{\text{ref}}}(\cdot \mid \mathbf{x}) \right) \right], \qquad (4)$$

where $\beta$ controls the optional KL regularization with the reference policy $p_{\theta_{\text{ref}}}$. For the aggregation prompts, we additionally sample $K$ candidates from $p_{\theta_{\text{ref}}}$ to construct the aggregation set $S_0$, resulting in the following objective:

$$\max_{\theta} \mathbb{E}_{(\mathbf{x},\mathbf{y})\sim\mathcal{D},S_0\sim p_{\theta_{\text{ref}}}(\cdot|\mathbf{x})} \left[ \mathbb{E}_{\tau\sim p_{\theta}(\cdot|\mathbf{x},S_0)} \left[ r(\tau, \mathbf{y}) \right] - \beta \, \text{KL} \left( p_{\theta}(\cdot \mid \mathbf{x}, S_0) \,\|\, p_{\theta_{\text{ref}}}(\cdot \mid \mathbf{x}, S_0) \right) \right]. \qquad (5)$$

This objective can be optimized using any off-the-shelf policy gradient algorithm, such as PPO (Ouyang et al., 2022), GRPO (Shao et al., 2024), or RLOO (Ahmadian et al., 2024), initializing $\theta$ with a copy of the base model parameters $\theta_{\text{ref}}$ or using a parameter-efficient fine-tuning technique. We use RLOO in all our experiments (§5.4) for its simplicity and good empirical performance.

## 5 EXPERIMENTS

We first demonstrate the effectiveness of RSA as a test-time scaling strategy in §5.1 and §5.2 through comprehensive evaluations on math, code generation, general reasoning, and knowledge recall

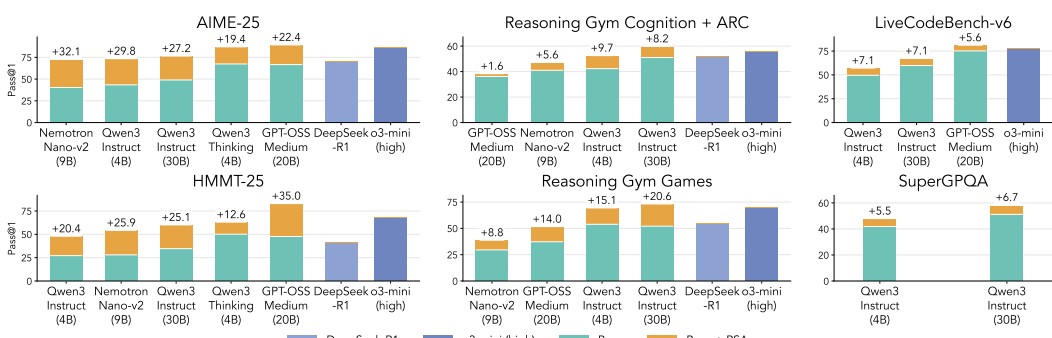

Figure 4: **RSA significantly improves Pass@1 across math, code, general reasoning, and knowledge recall tasks.** We observe consistent gains across diverse model families, including standard instruction-tuned models and long CoT "thinking" models. Further details provided in §E.2.

Table 1: We report Pass@1 scores for RSA and other test-time scaling baselines. RSA results obtained with aggregation size $K = 4$, population size $N = 16$, run for $T = 10$ steps. Majority-voting and rejection-sampling are budget-matched with RSA. Results are averaged over 4 seeds for all tasks except SuperGPQA, where we use 1 seed. Further details in §E.

| Method ↓ Task → | Math reasoning | | Code gen. | General reasoning / planning | | Knowledge recall |
| --- | --- | --- | --- | --- | --- | --- |
| | **AIME-25** | **HMMT-25** | **LiveCodeBench-v6** | **RG Games** | **RG Cognition+ARC** | **SuperGPQA** |
| Reference | 43.91 | 27.17 | 49.63 | 53.88 | 42.31 | 41.85 |
| Rejection sampling | 48.23 | 32.76 | 50.90 | 56.41 | 45.11 | 46.18 |
| Self-refinement | 53.33 | 39.17 | 51.40 | 65.50 | 49.17 | 43.5 |
| Majority voting | 68.33 | 35.00 | – | 65.15 | 46.07 | **48.2** |
| Self-aggregation (RSA, $T = 1$) | 56.51±3.06 | 36.15±1.26 | 51.94±0.15 | 65.21±0.98 | 48.63±0.81 | 45.91 |
| **RSA** | **73.18**±2.20 | **47.55**±1.00 | **56.72**±0.65 | **68.98**±1.10 | **51.96**±2.24 | 47.39 |

benchmarks. In §5.3, we analyze how RSA's three key parameters – the aggregation set size $K$, the population size $N$, and the number of sequential steps $T$ – contribute to its success. Finally, in §5.4, we show that aggregation-aware RL training can further enhance RSA's performance.

**Tasks.** We evaluate RSA across four benchmark categories, providing further details in §E.1:

- **Math.** We use AIME-25 and HMMT-25 from MathArena (Balunović et al., 2025), each containing 30 challenging competition-level math problems.
- **General reasoning.** We construct two datasets with 100 problems each from Reasoning Gym (Stojanovski et al., 2025), using tasks from the games category, and cognition + ARC categories.
- **Code generation.** We use LiveCodeBench-v6 (Jain et al., 2024) which contains 1055 problems.
- **Knowledge-based reasoning.** We use SuperGPQA (M-A-P Team et al., 2025), a graduate-level knowledge-based reasoning benchmark, to test effectiveness of RSA on tasks requiring factual recall. Given the large dataset size, we evaluate on 1000 randomly chosen multiple-choice questions.

### 5.1 RSA OUTPERFORMS OTHER TEST-TIME SCALING METHODS

We benchmark RSA against sequential and parallel test-time scaling methods with `Qwen3-4B-Instruct-2507` as the base model. To ensure consistency across tasks, we fix the population size to $N = 16$, the aggregation set size to $K = 4$, and the number of recursive updates to $T = 10$. Results are averaged over 4 seeds, except for SuperGPQA where we report a single seed due to computational constraints. For fairness, we restrict comparisons to methods that require no additional training or external verifiers. Further experimental details are provided in §E.3.

**Sequential baselines.** We consider $T$-step self-refinement (Madaan et al., 2023), which corresponds to RSA with $K = 1$ and $N = 1$, which we run for $T = 10$ steps.

**Parallel baselines.** We evaluate majority voting (Wang et al., 2023) and rejection sampling with self-verification (Weng et al., 2023), budget-matched with RSA by using $N \times T$ generations. We also include single-step self-aggregation (Li et al., 2025), equivalent to RSA with $K = 4$ and $T = 1$.

Table 1 reports Pass@1 results across all benchmarks, showing that RSA consistently outperforms both sequential and parallel baselines. Against self-refinement, RSA achieves higher performance, demonstrating that aggregating multiple solutions provides clear advantages over refining a single one. Notably, RSA with $T = 10$ outperforms its single-step variant ($T = 1$), highlighting the benefits

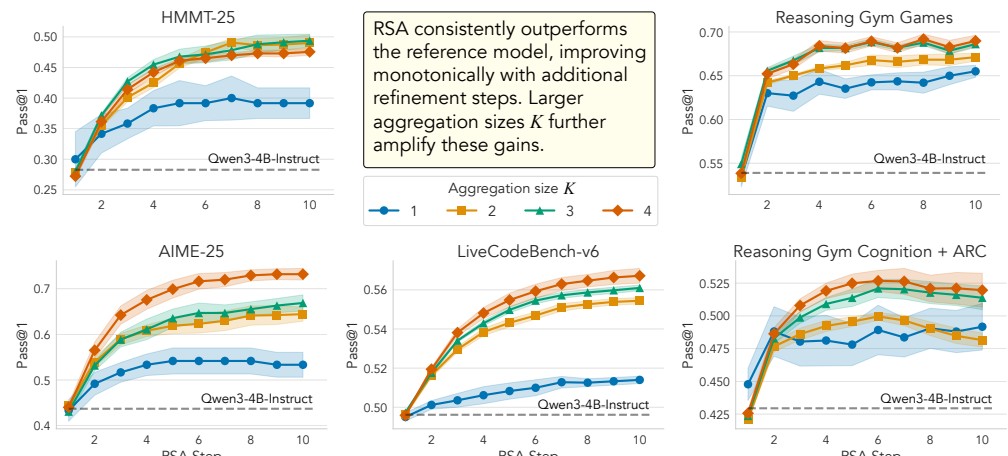

Figure 5: Pass@1 vs. RSA steps, for fixed population size $N = 16$, using `Qwen3-4B-Instruct-2507`. Error bands indicate standard deviation over 4 seeds. Larger $K$ generally improves performance.

of recursive aggregation. When compared to parallel methods, RSA achieves superior results on all tasks except SuperGPQA, where majority voting is particularly effective due to the multiple-choice answer format. We omit majority voting on LiveCodeBench-v6 since code solutions rarely coincide exactly, and refer to §E for further details.

## 5.2 RSA YIELDS CONSISTENT GAINS ACROSS DIFFERENT MODELS

We apply RSA to a diverse set of instruction-tuned models spanning a wide range of parameter counts, architectures, and reasoning abilities, including long chain-of-thought "thinking" models, sparse Mixture-of-Experts (MoE) architectures, and hybrid state-space models. Specifically, among instruction-tuned models, we consider `Qwen3-4B-Instruct-2507` and `Qwen3-30B-A3B-Instruct-2507` (Yang et al., 2025), while among long chain-of-thought thinking models, we include `Qwen3-4B-Thinking-2507`, `gpt-oss-20b (medium)` (OpenAI et al., 2025), and `NVIDIA-Nemotron-Nano-9B-v2` (NVIDIA, 2025). Table 4 provides a detailed overview of the characteristics of the models considered in this work.

Fig. 4 shows that RSA leads to substantial improvements on all tasks across a wide range of models. Remarkably, applying RSA to `Qwen3-4B-Instruct-2507`, a substantially weaker model, matches and in some cases outperforms strong reasoning models like `DeepSeek-R1` and `o3-mini (high)` without RSA. Taken together, these results establish RSA as a *strong and general* test-time scaling strategy for a wide variety of tasks.

## 5.3 EFFECT OF SCALING RSA HYPERPARAMETERS

We perform experiments using `Qwen3-4B-Instruct-2507` to answer the following questions:

- How does the performance of RSA vary with parallel and sequential scaling parameters?
- What underlying mechanisms explain the performance gains of RSA?
- How to select the parameters effectively under a compute budget?

**Monotonic improvement with sequential depth $T$.** Fig. 5 plots the Pass@1 scores over self-aggregation steps for different aggregation set sizes $K$. Performance improves monotonically on nearly all tasks, with the only significant downward trend on Reasoning Gym Cognition + ARC after five steps. Overall, these results demonstrate that RSA scales effectively with increasing depth.

**Increasing aggregation size $K$ improves performance.** Fig. 5 shows that increasing the aggregation size $K$ improves performance. The largest gain is observed when moving from $K = 1$ to $K = 2$, highlighting that aggregating over multiple reasoning chains provides substantial improvement over single-trajectory refinement. We observe diminishing returns beyond $K = 3$ on most tasks, possibly because the model cannot effectively attend to very long contexts.

**Effect of increasing population size $N$.** Next, we study the impact of total population size $N$, *i.e.*, the number of unique candidates available for aggregation at each step. In Fig. 6, we plot the final Pass@1 scores for different tasks using a fixed aggregation size of $K = 4$ and $T = 10$ sequential steps, while varying $N \in \{4, 8, 16, 32\}$. We observe that increasing $N$ always initially improves

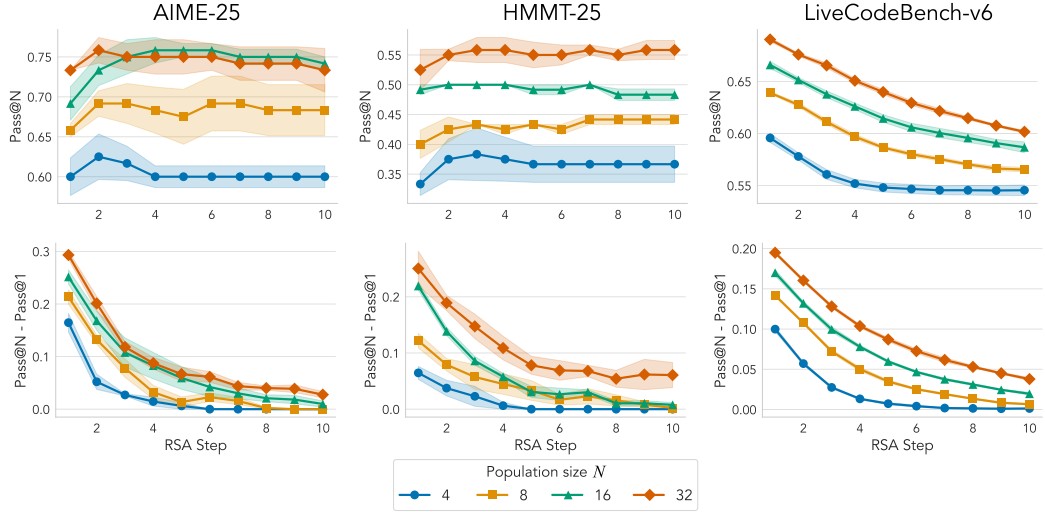

Figure 7: Pass@N (**top row**) and Pass@$N$ − Pass@1 (**bottom row**) across RSA steps for different values of $N$. Larger $N$ results in higher Pass@N score, but requires more steps to mix, delaying the convergence of Pass@1 to Pass@$N$. All results with fixed $K = 4$.

performance, but scaling $N$ to very large values leads to a small performance drop on AIME-25 and Reasoning Gym Games. We investigate the role of population size further in the following section, where it emerges as the key parameter controlling the asymptotic performance of RSA.

**Pass@N as a predictor of asymptotic performance.** The Pass@$N$ score for a population of $N$ solutions is equal to 1 if at least one final answer out of the $N$ is correct. The top row of Fig. 7 shows the average Pass@$N$ score of the population across iterations of RSA for different values of $N$. For the math tasks (AIME-25, HMMT-25), Pass@$N$ remains relatively stable, whereas for LiveCodeBench-v6 it decreases by 6-8%. As expected, larger $N$ yields a higher baseline Pass@$N$ score.

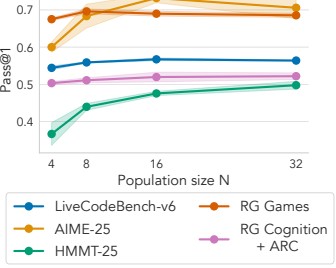

Figure 6: Pass@1 at $T = 10$ over population size $N$ (fixed $K = 4$).

We find that the gap between Pass@$N$ and Pass@1 is a useful predictor of the 'aggregability' of a set of solutions. The bottom row of Fig. 7 shows this gap over iterations. As the number of RSA iterations grows, Pass@1 converges to Pass@$N$, which acts as an upper bound on the performance. The Pass@$N$ − Pass@1 gap consistently drops faster for smaller $N$ with fixed aggregation set size $K$. Intuitively, RSA preserves good reasoning patterns in the population, and high-quality reasoning chains can mix within the population in fewer aggregation iterations if the population size is small. Therefore, a larger population size $N$ enables better asymptotic performance, but requires either more sequential iterations $T$ or faster mixing via larger aggregation size $K$. See §D for a population diversity analysis over RSA steps, which further validates these findings.

**Tuning RSA under compute budgets.** Our results indicate that jointly increasing $N$, $K$, and $T$ improves RSA performance. In practice, the key question is *how to scale them relative to one another given a limited compute budget*. Based on the above analysis we note that:

- Population size $N$ controls the asymptotic performance.
- Larger aggregation set size $K$ for a fixed $N$ leads to faster mixing of high quality chains (for $K > 1$).
- Longer self-aggregation depth $T$ monotonically improves performance.

When a higher number of sequential reasoning steps $T$ are feasible, it allows for a smaller $K$ provided $N$ is large. Conversely, when $T$ is limited due to time constraints and increasing $K$ is impractical (*e.g.*, due to context length constraints), $N$ should also be reduced; a large population that fails to mix effectively is less useful than a smaller batch that evolves rapidly together. We expect these findings to generalize to other model families.

### 5.4 RSA IMPROVES WITH AGGREGATION-AWARE RL

We next analyze the impact of the aggregation-aware RL training procedure described in §4.

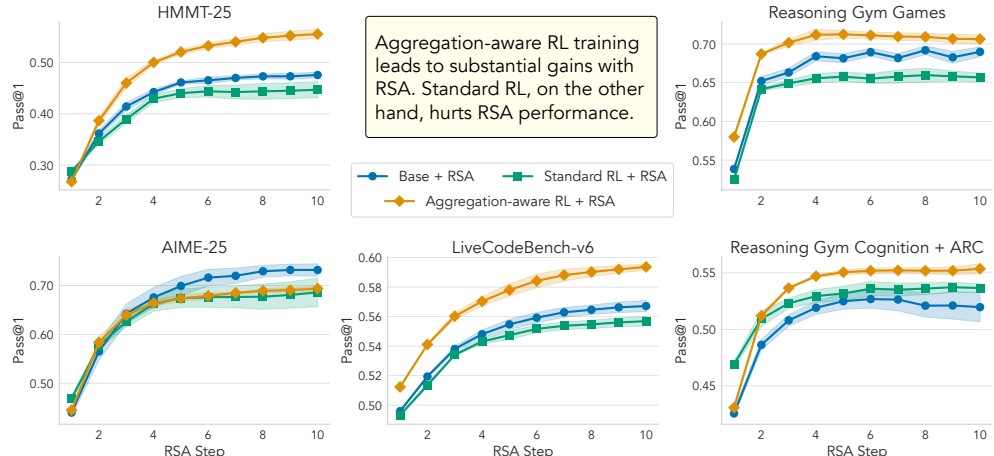

Figure 8: Pass@1 across RSA steps for the base, standard RL, and aggregation-aware RL policies with `Qwen3-4B-Instruct-2507`. Standard RL training generally hurts performance when using RSA, whereas aggregation-aware training leads to marked improvement on most tasks.

**Setup.** We use `Qwen3-4B-Instruct-2507` as the reference model. We construct a general reasoning dataset by combining 16,000 randomly sampled math problems from DeepScaleR (Luo et al., 2025), and 2048 problems each from six Reasoning Gym tasks where the reference model performs poorly (`tower_of_hanoi`, `sokoban`, `knight_swap`, `rush_hour`, `arc_1d`, and `sentence_reordering`). For each query in the dataset, we generate four candidate solutions using the reference model, which are then used to form aggregation prompts as shown in Fig. 3.

We train an *aggregation-aware* model on this augmented dataset by jointly optimizing (4) for the standard prompts and (5) for the aggregation prompts. As a baseline, we also train a model on the original dataset with only standard prompts by optimizing (4). The model trained using standard RL is only trained to generate solutions directly and is not optimized to aggregate reasoning chains. Both models are trained for 300 steps using RLOO (Ahmadian et al., 2024). (Further details in §E.4.) For evaluation, we run RSA for $T = 10$ steps with the reference, *standard RL* post-trained, and *aggregation-aware RL* post-trained models on AIME-25, HMMT-25, LiveCodeBench-v6, and the Reasoning Gym Games and Cognition + ARC test sets from §5. We ensure no data contamination between the training and test sets. We fix the aggregation size $K = 4$ and population size $N = 16$.

**Results.** Fig. 8 shows that in four out of five cases, RSA with the standard RL fine-tuned model underperforms RSA with the reference model, validating our hypothesis that distribution shifts incurred due to test-time scaling can lead to performance degradation after RL. The aggregation-aware policy, on the other hand, always outperforms standard RL and significantly outperforms the reference in four out of five tasks, with AIME-25 being the only outlier. Interestingly, we see massive gains on LiveCodeBench, despite the fact that our training dataset completely lacks any coding problems, which might indicate that the aggregation skills exhibit strong out of domain transferability. Overall, these experiments clearly demonstrate the benefits of aggregation-aware RL. Considering the implementation simplicity and the resulting robustness gains to RSA (or even to single-step self-aggregation), we strongly encourage its adoption for post-training.

## 6 CONCLUSION AND FUTURE WORK

We introduce Recursive Self-Aggregation (RSA), a hybrid test-time scaling framework that treats reasoning as an evolutionary process. By recursively aggregating reasoning chains, RSA enables models to cross-reference and recombine information across multiple candidates, while still retaining the depth of sequential refinement. This allows RSA to generate solutions that consistently outperform single-trajectory refinement and purely parallel scaling strategies. We further show that RL fine-tuning the LLM to perform aggregation amplifies RSA's benefits, yielding superior performance.

**Future work.** In future work, RSA can be composed with other test-time scaling methods to further improve performance, for example, by using self-verification to filter low-quality candidates from the population, thus introducing an explicit fitness function to the evolutionary algorithm. Another promising idea is to use multi-step reinforcement learning to train the policy for the end-to-end RSA procedure, moving beyond the greedy single-step aggregation explored in this work.

ETHICS STATEMENT

Improving test-time scaling performance of LLMs can enhance their usefulness across domains but also carries risks associated with misuse and unintended generalization. Careful evaluation and responsible deployment are essential as these methods are applied to more capable models.

REPRODUCIBILITY STATEMENT

We provide all the details to reproduce our results in §5 and §E. We also provide the code for our test-time scaling experiments at `https://anonymous.4open.science/r/RSA-85F6/README.md`.

LLM USE

LLMs were used to assist paper editing and to write the code for experiments.

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

# A   ADDITIONAL REBUTTAL EXPERIMENTS

We conduct new experiments as requested by the reviewers. In Table 2 we evaluate `Qwen3-4B-Instruct` using developer-recommended sampling parameters instead of the vllm defaults used for the other experiments. RSA is robust to the choice of sampling parameters, and we observe similar relative performance here as our main experiments. In Table 3, we include new experiments with a single-model implementation of the Mixture-of-Agents (Wang et al., 2024) method, which RSA significantly outperforms.

Table 2: Pass@1 scores averaged over 4 seeds (except LCB) with developer-recommended sampling parameters (temperature=0.7, TopP=0.8, TopK=20, MinP=0, max_response_len=16384). We use $K = 4, N = 16, T = 10$ for RSA, and rejection sampling and majority voting are budget-matched per-prompt with RSA.

| Method ↓ Task → | Math reasoning | | Code gen. | General reasoning / planning | | Knowledge recall |
|---|---|---|---|---|---|---|
| | AIME-25 | HMMT-25 | LiveCodeBench-v6 | RG Games | RG Cognition+ARC | SuperGPQA |
| Reference | 47.71 | 30.63 | 49.16 | 57.12 | 45.26 | 42.67 |
| Rejection sampling | 51.43 | 34.34 | 50.67 | 60.48 | 46.31 | 47.62 |
| Majority voting | 64.99 | 33.33 | – | 62.63 | 45.33 | **48.85** |
| **RSA** | **68.49** | **44.43** | **55.66** | **68.88** | **56.84** | 48.13 |

Table 3: Pass@1 scores averaged over 4 seeds with Qwen3-4B-Instruct. We use K=4, N=16, T=10 for RSA, and K=4, T=10 for self-MoA.

| Method ↓ Task → | AIME-25 | HMMT-25 |
|---|---|---|
| Reference | 43.91 | 27.17 |
| Majority voting | 68.33 | 35.0 |
| Self-MoA | 62.5 | 42.5 |
| **RSA** | **73.18** | **47.55** |

# B   ADDITIONAL RELATED WORK

**Chain-of-thought aggregation.** Several recent papers have explored self-aggregation as a parallel scaling strategy. Li et al. (2025) study simple single-step aggregation, while Wang et al. (2025) enhance aggregation ability through supervised fine-tuning (SFT), which requires access to a stronger teacher LLM. Concurrent to our work, Zhao et al. (2025) trained RL policies for single-step aggregation. Our work conducts more extensive experiments across a broader suite of tasks with ablations, and further motivate aggregation-aware RL as a means to improve the performance of test-time recursive aggregation as an additional contribution. Naik et al. (2023) does not use self-aggregation, instead including an "approach" in the prompt to generate multiple diverse solutions, from which an answer is selected using majority voting. The algorithm could easily be modified to use self-aggregation as the combination strategy instead. None of these works explored the sequential scaling and evolutionary components introduced in our work. Wang et al. (2024) is closely related to our approach, and uses an ensemble of LLMs to generate proposals that are jointly aggregated by a stronger model in an iterative loop. In contrast, RSA uses a single LLM and mixes the population by aggregating random subsets at each step while maintaining a fixed population size greater than the aggregation size to maintain population diversity, which we identify as critical factor (See §5.3 and §D for further analysis).

**Evolutionary methods.** Another line of work closely related to RSA is using LLMs within evolutionary algorithms. Yang et al. (2023) propose using LLMs as proposers and mutators within an evolutionary optimization loop. They assume access to an external fitness function to evaluate the solutions. Romera-Paredes et al. (2024) propose FunSearch which builds upon a similar idea using LLMs to modify and propose new python functions given a scoring function. Similar to our aggregation-aware RL approach, EvoTune (Surina et al., 2025) trains the LLM within an evolutionary process with RL to improve the LLM in the context of program synthesis.

**Other related works.** Several hybrid scaling strategies build on Tree-of-Thoughts (ToT) (Yao et al., 2023). (Inoue et al., 2025), expands trees over coherent text units ("thoughts") and applies adaptive branching to Monte Carlo Tree Search, with external or self-verification serving as the value function. Graph-of-Thoughts (GoT) (Besta et al., 2024) generalizes Tree-of-Thoughts by organizing reasoning units ("thoughts") into a directed acyclic graph (DAG), allowing nodes to have multiple parents (through aggregation) in addition to branching. Forest-of-Thought (Bi et al., 2025) expands

multiple ToTs in parallel, and combines their final solutions using majority voting; if no consensus is found, it uses an LLM to compare the reasoning processes and outcomes of the different trees to give a final answer. A key weakness of these approaches is their reliance on either external verification of outcomes or value functions for scoring partial reasoning chains, the latter being a notoriously difficult problem. They also typically require careful prompting to ensure that the generated chains consist of meaningful atomic "thoughts". To date, we are not aware of applications of these methods to long CoT reasoning models.

(Huang et al., 2024) found intrinsic self-correction capabilities in LLMs to be influenced by prompt-design. Our work indicates that aggregation prompts overcome pathologies with self-correction by injecting diversity into the prompt. (Chen et al., 2025) is a similar strategy that overcomes the mode collapse problem with self-refinement by maintaining $N$ parallel chains, with the answer finally extracted using majority-voting. Unlike RSA, these chains remain completely independent across the sequential steps, so the diversity is highly dependent on the initial proposal generation.

## C  RSA ALGORITHM

---

**Algorithm 1:** Recursive Self-Aggregation (RSA)

---

**Input:** LLM $p_\theta$ with fixed parameters, problem $\mathbf{x}$, population size $N$, subset size $K$, steps $T$.
**Output:** Final population $\mathcal{P}_T = \{\tau_1^{(T)}, \ldots, \tau_N^{(T)}\}$.

// Initialization
1  $\{\tau_i^{(1)}\}_{i=1}^N \sim p_\theta(\cdot \mid \mathbf{x})$
2  $\mathcal{P}_1 \leftarrow \{\tau_1^{(1)}, \ldots, \tau_N^{(1)}\}$

// Recursively for $t = 1, \ldots, T - 1$
3  **for** $t \leftarrow 1$ **to** $T - 1$ **do**
    // Subsampling
4   **for** $i \leftarrow 1$ **to** $N$ **do**
5     Sample $\{m_{i,1}^{(t)}, \ldots, m_{i,K}^{(t)}\} \sim \text{Uniform}(\{1, \ldots, N\})$ indices without replacement
6     Form aggregation set $S_i^{(t)} \leftarrow \{\tau_{m_{i,1}^{(t)}}^{(t)}, \ldots, \tau_{m_{i,K}^{(t)}}^{(t)}\}$
7    $\mathcal{S}_t \leftarrow \{S_1^{(t)}, \ldots, S_N^{(t)}\}$

    // Self-aggregation
8   **for** $i \leftarrow 1$ **to** $N$ **do**
9     $\tau_i^{(t+1)} \sim p_\theta(\cdot \mid S_i^{(t)}, \mathbf{x})$
10   $\mathcal{P}_{t+1} \leftarrow \{\tau_1^{(t+1)}, \ldots, \tau_N^{(t+1)}\}$

// Termination
11 **return** $\mathcal{P}_T$ **and optionally** sample $\hat{\tau} \sim \text{Uniform}(\mathcal{P}_T)$.

---

## D  POPULATION DIVERSITY ANALYSIS

**Setup.**  In this experiment, we study how the diversity of the population evolves across RSA steps, using the AIME-25 dataset as an illustrative example. To quantify this, we require a metric to measure semantic diversity within a batch of CoTs. We embed the reasoning chains to generate sequence embeddings with ModernBERT (Warner et al., 2024), a strong Transformer encoder, and use the average cosine distance between embeddings in the population as a simple diversity metric. While this is an imperfect metric, it can still reveal interesting trends when plotted over time.

**Effect of varying $K$.**  The left of Fig. 9 plots the average population diversity across RSA steps with a fixed population size $N = 16$ and varying the aggregation size $K = 2, 3, 4$. For all $K$, diversity rises sharply after the first aggregation step ($t = 2$) and then steadily decays. After 10 steps, larger $K$ yields lower diversity. This aligns with our intuition and previously observed results – larger aggregation sizes promote faster mixing of high-quality samples, leading to quicker convergence toward high-reward solutions (Fig. 5). Conversely, smaller $K$ slows mixing, explaining the weaker performance observed in Fig. 5.

**Effect of varying** $N$. The right panel of Fig. 9 shows average population diversity across RSA steps with fixed aggregation size $K = 4$ and varying population sizes $N = 4, 8, 16$. After 10 steps, diversity is lowest for $N = 4$ and highest for $N = 16$, though the relative differences are smaller than in the previous experiment varying $K$. Taken together with the earlier result that larger $K$ accelerates mixing and improves performance for fixed $N$, these findings support our hypothesis from §5.3 – scaling up $N$ should be accompanied by increasing $K$ or $T$, otherwise the batch will fail to mix in time and the average sample quality will remain poor after $T$ steps.

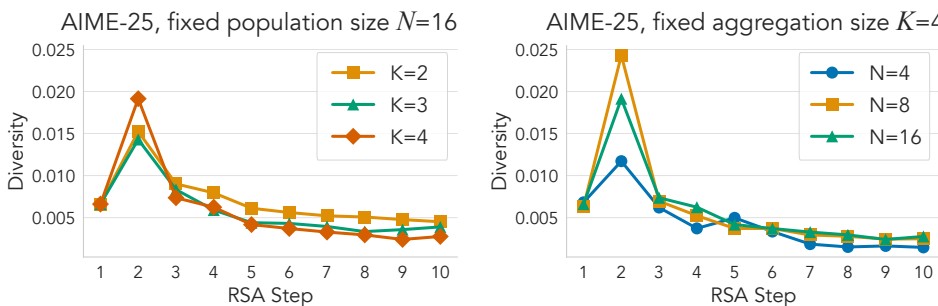

Figure 9: **Left:** Diversity over RSA steps with fixed population size $N = 16$ and varying aggregation batch $K$ on AIME-25. Larger $K$ accelerates mixing, shown through faster drop in population diversity. **Right:** Diversity over RSA steps with fixed $K = 4$ and varying $N$ on AIME-25. Increasing $N$ enhances the diversity of reasoning chains, and hence the Pass@$N$ score, which determines asymptotic performance. However, very large $N$ relative to $K$ can slow mixing and hinder performance.

## E   EXPERIMENT DETAILS

In this section we provide all experiment details necessary to reproduce our results. We provide code here.

### E.1   TASK DETAILS

- **Math.** We evaluate on the complete AIME-25 and HMMT-25 datasets, each consisting of 30 problems. These tasks use a binary reward: 1.0 if the predicted answer is symbolically equivalent to the ground truth and 0.0 otherwise, evaluated using Math-Verify.
- **General reasoning.** Reasoning Gym (Stojanovski et al., 2025) consists of a broad suite of tasks divided into different categories, and difficulty levels. For our evaluations, we construct two datasets – one from the "Games" category (17 tasks), requiring general reasoning and planning, and the other by combining the "Cognition" and "ARC" categories (7 + 2 tasks), requiring pattern recognition. Each of the datasets consists of 100 randomly generated problems, equally split between the tasks in the categories. We selected these categories as we found them to be the most difficult for our base models, particularly `Qwen3-4B-Instruct-2507` which we use for our ablations. We evaluate on the "easy" version of the problems, since the "hard" version is significantly more challenging, with even frontier reasoning models obtaining 0.0 reward on most tasks. The reward function is task-dependent, and can be found alongside task descriptions in the Reasoning Gym repository.
- **Code generation.** We evaluate code generation using the complete LiveCodeBench-v6 dataset (Jain et al., 2024) consisting of 1055 problems. The task uses a binary reward of 1.0 if the generated Python code passes all provided test cases upon execution, else 0.0.
- **Knowledge based reasoning.** We use SuperGPQA (M-A-P Team et al., 2025) as our knowledge-based reasoning benchmark. Although RSA is designed to enhance deep reasoning, we include this dataset for completeness and still observe substantial gains. SuperGPQA consists of multiple-choice questions and assigns a binary reward of 1.0 when the selected option matches the ground truth.

### E.2   MODELS USED

We use `Qwen3-4B-Instruct-2507` as the core model for all experiments and ablations. This choice was motivated by its small parameter count, which makes inference and RL fine-tuning tractable, while still offering strong base reasoning ability. For all models, we fix the response length to values

Table 4: List of models used in our experiments. We include Mixture-of-Experts (MoE) (Shazeer et al., 2017) and state-space model (SSM) architectures (Gu et al., 2022; Gu & Dao, 2024). Our evaluation spans both non-thinking and thinking models, with appropriately larger response lengths allocated to the latter.

| Model | MoE | Hybrid-SSM | Thinking | Response Length |
|---|---|---|---|---|
| Qwen3-4B-Instruct-2507 | ✗ | ✗ | ✗ | 8192 |
| Qwen3-30B-A3B-Instruct-2507 | ✓ | ✗ | ✗ | 8192 |
| Qwen3-4B-Thinking-2507 | ✗ | ✗ | ✓ | 32768 |
| gpt-oss-20b (medium) | ✓ | ✗ | ✓ | 16384 |
| NVIDIA-Nemotron-Nano-9B-v2 | ✗ | ✓ | ✓ | 16384 |

that avoid frequent truncation and keep this constant across tasks. All model details, including their characteristics and response lengths, are tabulated in Table 4.

### E.3 Baseline details

- **Rejection sampling.** We prompt the model to self-verify $N = 160$ candidate solutions. We then compute the mean score over the positively sampled solutions – equivalent in expectation and lower variance than sampling one of these at random.
- **Self-refinement.** For $T = 10$ steps, generated solutions are fed back into the model, which is prompted to detect errors and refine its reasoning chain.
- **Majority voting.** We extract the final answers from all $N = 160$ reasoning chains and group equivalent ones. The majority group is then selected and compared with the ground truth.
- **Self-aggregation.** Equivalent to RSA with a single step of aggregation, we first generate a batch of $K = 4$ solutions and aggregate them with the model to produce the final answer. We stick to $K = 4$ for self-aggregation since the context lengths cannot scale beyond this point without model performance degradation. As a result, this baseline is not "budget-matched" with RSA similar to the other parallel baselines above, but we note that the ability to grow the effective batch size well above the context constraints of the model is one of the major advantages of recursive aggregation over single-step aggregation.

**Budget matching.** The budget matching is done per-prompt. Each prompt with RSA is processed with $N = 16$ total generations per step, for $T = 10$ steps. This means a total of 160 reasoning trajectories are sampled per prompt. To match this, we run majority voting, rejection-sampling and self-aggregation baselines using 160 generations, each generation limited to the same max response length used by RSA. We acknowledge that the actual responses do not have to fully use this budget, but the same is true for RSA, which may converge in fewer steps. Exact token parity isn't possible without forced sequence extension and truncation, which hurts performance without fine-tuning.

### E.4 RL training setup

We use `verl` (Sheng et al., 2024) as our framework to train the RL policies described in §5.4. The aggregation-aware dataset is a 50-50 split between standard and aggregation prompts. To generate the aggregation prompts, we use our standard inference procedure §E.5 with the `Qwen3-4B-Instruct-2507` reference policy to generate $K = 4$ candidate reasoning chains per query. All RL training parameters are shared between the standard and aggregation-aware RL training runs. We use RLOO (Ahmadian et al., 2024) as the training algorithm with the following hyperparameters; learning rate $= 1e - 6$, KL coefficient $\beta = 0.0$, batch size $= 256$, training steps $= 300$, response length $= 8192$, and max prompt length $= 33792$ to fit the aggregation prompts.

### E.5 LLM inference settings

For consistency and fairness, we share the same inference settings across all experiments. We use vLLM (Kwon et al., 2023) for both RSA and all baselines. We keep sampling parameters consistent across experiments. We set the sampling `temperature = 1.0`, `top_p = 1.0`, and `min_p = 1.0`, all of which are the default settings.

## F  QUALITATIVE EXAMPLE

We present a qualitative example below, where we provide `Qwen3-4B-Instruct-2507` with four candidate solutions for the problem: "Compute the sum of the positive divisors (including 1) of 9! that have units digit 1."

We highlight reasoning steps in the aggregated solution that are lifted from individual candidates, including parts that are newly added by the model and were not present in any candidate. We also provide the relevant parts from the candidate solutions that appear in the aggregated solution.

- **Step 1:** While all candidate solutions begin by calculating 9!, some of them compute the products pairwise. The language in the aggregated solution most closely mirrors the first candidate .
- **Step 2:** Different candidates compute prime factors in slightly different ways. The only solution that mentions Legendre's formula explicitly is the second candidate .
- **Step 3:** All solutions identified that any multiple of 5 cannot have a units digit of 1. However, only the fourth candidate correctly identified that even numbers also cannot have a units digit of 1, which is then used in the aggregated solution. This significantly shortens the search space for valid divisors.
- **Step 4:** This step had the most diversity among the candidates. The first candidate exhaustively listed all possible combinations and pooled them into six different cases. While the aggregated solution considers the same divisors as candidate #4 (because of correctly identifying it can't be an even number), it arranges them into a table, which was not done in any individual candidate. This shows that the aggregated solution can add new information not present in any candidates.

---

**Aggregated Solution**

---

We are asked to compute the sum of the positive divisors of 9! that have a **units digit of 1**.

---

**Step 1: Compute** 9!

$$9! = 9 \times 8 \times 7 \times 6 \times 5 \times 4 \times 3 \times 2 \times 1 = 362880$$

---

**Step 2: Prime Factorization of** 9!

Using Legendre's formula, the prime factorization is:

$$\text{Exponent of 2: } \left\lfloor \frac{9}{2} \right\rfloor + \left\lfloor \frac{9}{4} \right\rfloor + \left\lfloor \frac{9}{8} \right\rfloor = 4 + 2 + 1 = 7$$

$$\text{Exponent of 3: } \left\lfloor \frac{9}{3} \right\rfloor + \left\lfloor \frac{9}{9} \right\rfloor = 3 + 1 = 4$$

$$\text{Exponent of 5: } \left\lfloor \frac{9}{5} \right\rfloor = 1 \qquad \text{Exponent of 7: } \left\lfloor \frac{9}{7} \right\rfloor = 1$$

Thus,

$$9! = 2^7 \times 3^4 \times 5^1 \times 7^1$$

---

**Step 3: Identify Divisors with Units Digit 1**

We are to find all **positive divisors** $d$ of 9! such that $d \bmod 10 = 1$.

Note:

- Any divisor divisible by **5** must end in **0 or 5**. Thus, **no divisor divisible by 5** can have units digit 1.

- Any divisor divisible by **2** is even → ends in an even digit (0, 2, 4, 6, 8).

  So **no even divisor** can end in 1.

Therefore, any divisor ending in 1 must be:

- **Odd** → not divisible by 2 or 5

---

- **Not divisible by 5**

- So must be a product of powers of **3 and 7 only**

That is, the divisor must be of the form:

$$d = 3^b \cdot 7^c$$

where:

$$0 \le b \le 4 \quad (\text{since } 3^4 \mid 9!), \qquad c \in \{0, 1\} \quad (\text{since } 7^1 \mid 9!).$$

**Step 4: List All Such Divisors with Units Digit 1**

We compute all $d = 3^b \cdot 7^c$ for $b = 0, \ldots, 4$, $c \in \{0, 1\}$, and check which end in 1.

| $b$ | $3^b$ | $c$ | $d$ | Units Digit |
|---|---|---|---|---|
| 0 | 1 | 0 | 1 | 1 ✓ |
| 0 | 1 | 1 | 7 | 7 ✗ |
| 1 | 3 | 0 | 3 | 3 ✗ |
| 1 | 3 | 1 | 21 | 1 ✓ |
| 2 | 9 | 0 | 9 | 9 ✗ |
| 2 | 9 | 1 | 63 | 3 ✗ |
| 3 | 27 | 0 | 27 | 7 ✗ |
| 3 | 27 | 1 | 189 | 9 ✗ |
| 4 | 81 | 0 | 81 | 1 ✓ |
| 4 | 81 | 1 | 567 | 7 ✗ |

Only the following values end in 1:

$$1, \quad 21, \quad 81.$$

All are divisors of 9!:

$$1: \text{trivial divisor}, \qquad 21 = 3 \times 7: \text{both powers present}, \qquad 81 = 3^4: 3^4 \mid 9!.$$

**Step 5: Compute the Sum**

$$1 + 21 + 81 = \boxed{103}$$

**Final Answer**

$$\boxed{103}$$

---

**Candidate #1**

We are asked to compute the **sum of the positive divisors of 9**! that have **units digit 1**.

**Step 1: Compute 9**!

$$9! = 9 \times 8 \times 7 \times 6 \times 5 \times 4 \times 3 \times 2 \times 1$$

We can compute it step by step:

- $9! = 362880$

$\vdots$

---

**Candidate #2**

We are asked to compute the **sum of the positive divisors of 9**! that **end in the digit 1** (*i.e.*, have **units digit 1**).

$\vdots$

---

**Step 2: Prime factorization of 9!**

We find the prime factorization of 9! using Legendre's formula:

For a prime $p$, exponent in $n!$ is:

$$\sum_{k=1}^{\infty} \left\lfloor \frac{n}{p^k} \right\rfloor$$

Primes $\leq 9$: 2, 3, 5, 7

**Exponent of 2:**

$$\left\lfloor \frac{9}{2} \right\rfloor + \left\lfloor \frac{9}{4} \right\rfloor + \left\lfloor \frac{9}{8} \right\rfloor = 4 + 2 + 1 = 7$$

**Exponent of 3:**

$$\left\lfloor \frac{9}{3} \right\rfloor + \left\lfloor \frac{9}{9} \right\rfloor = 3 + 1 = 4$$

**Exponent of 5:**

$$\left\lfloor \frac{9}{5} \right\rfloor = 1$$

**Exponent of 7:**

$$\left\lfloor \frac{9}{7} \right\rfloor = 1$$

So,

$$9! = 2^7 \times 3^4 \times 5^1 \times 7^1$$

---

$\vdots$

---

## Candidate #4

We are asked to compute the **sum of the positive divisors of 9!** that have **units digit 1**.

$\vdots$

---

**Step 3: List all divisors of** 9! **that end in 1**

Rather than list *all* divisors (which would be $(7 + 1)(4 + 1)(1 + 1)(1 + 1) = 8 \times 5 \times 2 \times 2 = 160$), we need only the ones ending in **1**.

We can **generate the divisors** in a smart way, or use modular arithmetic to filter.

But since the total number is 160, and many divisors end in 0, 2, 4, 5, 6, 8, we can hope that only a few end in 1.

Let's consider the **last digit** of a divisor.

We can iterate through all divisors and check the last digit, but we can also use **modular filtering**:

We are to find all $d \mid 9!$ such that $d \equiv 1 \pmod{10}$

Note: If $d \mid 9!$, and $d \equiv 1 \pmod{10}$, then $d$ is **odd** (since it ends in 1), and $d \not\equiv 0 \pmod 5$, because any multiple of 5 ends in 0 or 5, and only 5 ends in 5, but 1 is not 5.

So **d cannot be divisible by 5**.

That's a key observation!

**Key Observation:**

- Any divisor of 9! ending in 1 must **not** be divisible by 5.

- Because: If divisible by 5 → ends in 0 or 5 → can't end in 1.

So, we are only interested in **divisors of** 9! **that are not divisible by 5** and **end in 1**.

But even among those not divisible by 5, most end in 2, 4, 6, 8 — only some end in 1.

So actual candidate divisors: divisors of $\frac{9!}{5} = \frac{362880}{5} = 72576$

---

Since we are excluding all divisors divisible by 5.
So, we now consider only **divisors of** $2^7 \times 3^4$ (since we remove the 5 factor), and among them, those ending in **1**.
So the possible divisors (that are not divisible by 5) are all numbers of the form:

$$d = 2^a \times 3^b \quad \text{where } 0 \le a \le 7, 0 \le b \le 4$$

We want those such that $d \equiv 1 \pmod{10}$
Let's go through all combinations of $(a, b)$, compute $d = 2^a \times 3^b$, and check if $d \mod 10 = 1$

$\vdots$

Wait — **important point:**

- If $a \ge 1$, then $d$ is even $\rightarrow$ last digit is even $\rightarrow$ cannot be 1.
- So **only when** $a = 0$ can a divisor $d$ be odd, hence end in 1.

Thus, **all divisors ending in 1 must have** $a = 0$.

$\vdots$

What about **7**?
Ah! We forgot about 7!
We initially focused on $2^a \cdot 3^b$, but we removed 5, but **we haven't included the factor of 7**!
Wait — this is a **major mistake**.
We said "divisors not divisible by 5" — thatś correct.
But divisors can include **7** — since $7^1$ divides 9!

$\vdots$

# G   RSA PROMPTS

We use a simple prompt template with minor task-appropriate changes, without tuning beyond some initial adjustments. Although greater gains may be achievable with careful prompt engineering, we avoided this to prevent skewing results, and instead report performance that can be reasonably expected from a straightforward implementation of RSA. In fact, we look forward to future work that applies automated prompt optimization, such as GEPA (Agrawal et al., 2025), to design prompts tailored to the end-to-end RSA procedure, offering a cheaper alternative to the RL-based approach discussed at the end of §6. The prompt generation functions for math (AIME-25, HMMT-25), Reasoning Gym, and SuperGPQA tasks are given below. The prompt illustrated in Fig. 3 is the essentially the same prompt without any task-specific formatting. The prompts used for LiveCodeBench are similar, but contain some additional instructions to condition on the provided starter code.

**RSA prompt generation code**

```python
def aggregate_prompt(query: str, candidate_answers: List[str], task: str) -> str:
    # Reasoning Gym
    if task == 'rg':
        problem_kind = 'problem'
        format_hint = '<answer>...</answer>'
    # SuperGPQA
    elif task == 'supergpqa':
        problem_kind = 'multiple-choice problem'
        format_hint = '\\boxed{}. Only include the correct option letter in \\boxed{}; for example \\
      boxed{A}'
    # Math (AIME-25, HMMT-25)
    else:
        problem_kind = 'math problem'
        format_hint = '\\boxed{}'

    parts = []
    # If K=1, we are doing single-trajectory refinement
    if len(candidate_answers) == 1:
        parts.append(
            f"You are given a {problem_kind} and a candidate solution. "
            "The candidate may be incomplete or contain errors. "
            "Refine this trajectory and produce an improved, higher-quality solution. "
            "If it is entirely wrong, attempt a new strategy. "
            f"End with the final result in {format_hint}.\n"
```

```
24              )
25         # If K>1, we are doing multi-trajectory aggregation
26         else:
27              parts.append(
28                  f"You are given a {problem_kind} and several candidate solutions. "
29                  "Some candidates may be incorrect or contain errors. "
30                  "Aggregate the useful ideas and produce a single, high-quality solution. "
31                  "Reason carefully; if candidates disagree, choose the correct path. "
32                  "If all are incorrect, then attempt a different strategy."
33                  f"End with the final result in {format_hint}.\n"
34              )
35
36         parts.append("Problem:\n")
37         # Original query is appended here.
38         # It also contains the formatting instructions.
39         parts.append(query.strip() + "\n")
40
41         # If K=1, we are doing single-trajectory refinement
42         if len(candidate_answers) == 1:
43              parts.append("Candidate solution (may contain mistakes):\n")
44              ans_str = (candidate_answers[0] or "").strip()
45              parts.append(f"---- Candidate ----\n{ans_str}\n")
46              parts.append(
47                  f"Now refine the candidate into an improved solution. "
48                  "Provide clear reasoning and end with the final answer in {format_hint}."
49              )
50         # If K>1, we are doing multi-trajectory aggregation
51         else:
52              parts.append("Candidate solutions (may contain mistakes):\n")
53              for i, ans in enumerate(candidate_answers, 1):
54                  ans_str = (ans or "").strip()
55                  parts.append(f"---- Solution {i} ----\n{ans_str}\n")
56              parts.append(
57                  f"Now write a single improved solution. Provide clear reasoning and end with the final answer
          in {format_hint}."
58              )
59
60         return "\n".join(parts)
61
62  def build_prompt(query: str, candidate_answers: Optional[List[str]], task: str):
63         # At t=1, candidate_answers is None and we return base query
64         # At t>1, candidate_answers is a list of strings and we return an aggregated prompt
65         if candidate_answers is not None:
66              prompt = aggregate_prompt(query, candidate_answers, task)
67         else:
68              prompt = query
69         return prompt
```

