# OpenReview forum: "Recursive Self-Aggregation Unlocks Deep Thinking in Large Language Models"
_ICLR.cc/2026/Conference — Submitted to ICLR 2026_

### Official Review · Reviewer_Jzh2 · 2025-10-26

**Soundness:** 2
**Presentation:** 3
**Contribution:** 2
**Rating:** 4
**Confidence:** 4

**Summary:**

The paper introduce RSA, a novel test-time scaling algorithm which maintain a set of response and keep improving through self-aggregation on different subsets. The proposed method are evaluated on different models and different datasets to show its effectiveness, and further show how such approach can be intergrated to aggregation aware-RL to improve make the RL more effective.

**Strengths:**

- The paper is well-written and easy to follow.
- The experiments are comprehensive, including many SOTA models on different datasets. A detailed ablation study is provided to support the parameter choice of the method and provide a recommendation on more general cases.
- The experiments on integrating RSA to RL are very insightful.

**Weaknesses:**

- While the experiments have included many models, one concern is that using self-aggregation on non-thinking models provides an unfair benefit to make the model think longer. Similar concerns applies on reasoning models as the response length is capped. While there are still a decent number of models left for the main results, ablation study and further discussions like aggregation-aware RL lack such support.
- The sampling parameter used is relatively weird, and making the reported numbers far away from typical performance of the models. For example, the reported AIME and HMMT numbers are around 4% lower than the official report, while Livecodebench v6 is significantly higher (14% more). Similar difference on numbers also applies to results in Fig. 4 even if the exact numbers are not provided. While I personally think the idea could still provide a certain amount of benefit, given the fact that most researchers and users will adopt the recommended parameters of the models, I would recommend the authors to rerun the experiments to align to the recommended parameters, i.e., _Temperature=0.7, TopP=0.8, TopK=20, and MinP=0_ for Qwen and _temperature to 0.6, top_p to 0.95_ for nematron.
- How is the budget-match actually conducted? Is it per-prompt? Or per-dataset?

**Questions:**

No specific questions. Please refer to the weakness. Specifically, I would expect some rerun on the experiments to make the numbers match the normal results of the models on the datasets.

---

> ### Author Response · Authors · 2025-11-17
>
> We thank the reviewer for their feedback and hope that our response addresses all the concerns raised.
>
> > The sampling parameter used is relatively weird, and making the reported numbers far away from typical performance of the models. For example, the reported AIME and HMMT numbers are around 4% lower than the official report, while Livecodebench v6 is significantly higher (14% more). Similar difference on numbers also applies to results in Fig. 4 even if the exact numbers are not provided. While I personally think the idea could still provide a certain amount of benefit, given the fact that most researchers and users will adopt the recommended parameters of the models, I would recommend the authors to rerun the experiments to align to the recommended parameters, i.e., Temperature=0.7, TopP=0.8, TopK=20, and MinP=0 for Qwen and temperature to 0.6, top_p to 0.95 for nematron.
>
> Since our experiments span multiple model families and a diverse suite of tasks, we use the vLLM default sampling parameters (such as temperature 1), for consistency across models and tasks. Further, our parameters describe sampling according to the "default" distribution modeled by the LLM after pretraining. We believe that this is a fair setting to study test-time scaling, especially since optimal sampling parameters for the reference policy used in the standard way may not remain optimal after test-time scaling or RL post-training. Of course, we acknowledge that tuning these hyperparameters could further improve performance, but by keeping parameters consistent across models, tasks, and baselines we are able to isolate the performance gains due to RSA.
>
> Having said that, we agree with the reviewer that some experiments with the recommended sampling parameters can lend further credibility to our results. Following the reviewer's suggestion, we ran experiments on AIME-25, HMMT-25 and LiveCodeBench-v6 (LCB) with Qwen3-4B-Instruct-2507 using the recommended parameters. (We note that it would be quite expensive to rerun all the experiments with different samping parameters in the limited rebuttal period, so we focus on a few tasks to identify trends.)
>
> The reference scores below match the ones in the official report for AIME and HMMT. For LCB, we run a single seed due to compute limitations. The slightly lower (3-4%) scores of RSA and majority voting can be explained by the reduced diversity owing to these sampling parameters, especially lower temperature. We see an improved reference performance on LCB because we do not pass LCB's default `###` as a stop string to halt generation. When we run the base Qwen3-4B-Instruct-2507 model with this stop string, it leads to a success rate of 32.77. However, for all our experiments we don't add `###` as a stop string to prevent premature cutoff of model's responses.
>
> *Pass@1 scores averaged over 4 seeds (except LCB). We use $K=4, N=16, T=10$ for RSA, and rejection sampling and majority voting are budget-matched per-prompt with RSA.*
> | Method | AIME-25 | HMMT-25 | LCB-v6 |
> |-----------|----------:|----------:|----------:|
> | Reference |    47.71       |     30.63      |     49.16      |
> | Rejection Sampling    |    51.43       |     34.34      |     50.67      |
> | Majority voting    |     64.99      |     33.33      |     N/A      |
> | **RSA**   |    **68.49**       |    **44.43**       |    **55.66**       |
>
> **The general trends are consistent with our main results: with the recommended sampling parameters, RSA still significantly outperforms the baselines.**
>
> (continued in comment below)

---

> ### Author Response · Authors · 2025-11-17
>
> (continued from comment above)
>
>
> > While the experiments have included many models, one concern is that using self-aggregation on non-thinking models provides an unfair benefit to make the model think longer. Similar concerns applies on reasoning models as the response length is capped. While there are still a decent number of models left for the main results, ablation study and further discussions like aggregation-aware RL lack such support.
>
> We interpret the reviewer's concern as "The maximum response length might unfairly constrain thinking models, which need more output length compared to non-thinking models". If this was not the intended meaning, we politely ask the reviewer to clarify their concern and we will be happy to address it.
>
> We used a response length that allows the models to complete their chains of thought and output a final answer for most problems. We used the same response length (among other sampling parameters) and match compute per prompt for all baseline methods in Table 1 for a fair comparison. We direct the reviewer to the response lengths listed for each model in Table 2 (Appendix D), which were chosen to minimize truncations with the standardized sampling parameters for both thinking and non-thinking models. Indeed, a motivation of RSA is to facilitate "deep thinking", which cannot be achieved through simply scaling maximum response lengths.
>
> > How is the budget-match actually conducted? Is it per-prompt? Or per-dataset?
>
> The budget matching is done per-prompt. Each prompt with RSA is processed with $N=16$ generations per step, for $T=10$ steps. This means a total of 160 reasoning trajectories are sampled per prompt. We then run majority voting and other baselines with 160 generations, each one with the same max response length as used by RSA. We acknowledge that the actual responses do not fully use this budget, but the same is true for RSA, which may converge in fewer steps. Exact token parity isn't possible without forced sequence extension and truncation, which hurts performance without fine-tuning.
>
> **We hope we have addressed your concerns, especially regarding the sampling parameters, for which we have performed a new experiment. If you feel we have done so, please consider updating the review. We are happy to engage further if you have additional questions.**

---

> > ### Comment · Reviewer_Jzh2 · 2025-11-23
> >
> > I thank the authors for the rebuttal.
> >
> > I am relatively satisfied with the response on budget matching, and I do encourage the authors to update the paper draft immediately to add these clarifications to the paper. However, one concern is that aggregation also requires an LLM to finish, so theoretically, majority voting should get an even bigger number.
> >
> > The updated experiments are definitely helpful. However, given the vast difference in the overall performance on all other models and benchmarks, I am not able to update any score unless most scores are updated and match the typical performance. While I understand the computational constraint, I would recommend that the authors use the remaining time of the rebuttal to choose a more meaningful subset of the results and leave the remaining results in the appendix.
> >
> > Regarding the question related to response length, unfortunately, the interpretation is not exactly my concern. I am concerned about the fact that RSA is simply forcing non-thinking models to think more, whose main idea is the same as previous work, like self-verify and self-correct [1, 2, 3]. Please provide a discussion and comparison to these methods.
> >
> > [1]. Weng Y, Zhu M, Xia F, et al. Large language models are better reasoners with self-verification[C]//Findings of the Association for Computational Linguistics: EMNLP 2023. 2023: 2550-2575.
> >
> > [2]. Huang, Jie, et al. "Large language models cannot self-correct reasoning yet." arXiv preprint arXiv:2310.01798 (2023).
> >
> > [3]. Chen J, Ren J, Chen X, et al. Sets: Leveraging self-verification and self-correction for improved test-time scaling[J]. arXiv preprint arXiv:2501.19306, 2025.

---

> > > ### Author Response · Authors · 2025-11-25
> > >
> > > We thank the reviewer for engaging in discussion.
> > >
> > > > encourage the authors to update the paper draft
> > >
> > > We have added this to Appendix E.3 (all rebuttal additions use red text).
> > >
> > > > aggregation also requires an LLM to finish
> > >
> > > RSA ***does not require an LLM to finish generation*** for aggregation (Line 224). RSA operates under **exactly the same constraints** as our baselines, with the same sampling parameters and response length. We view this as a strength of RSA that it can bootstrap even from partial responses.
> > >
> > > **Impact of longer response length**: Additionally, we note that our experiments with the developer recommended parameters in the previous response are with a longer response length (=16k) where there were no cutoffs, and RSA still has a similar relative improvement over baselines.
> > >
> > > We have clarified both of these points in the revised draft.
> > >
> > > > However, given the vast difference in the overall performance on all other models and benchmarks, I am not able to update any score unless most scores are updated and match the typical performance.
> > >
> > > We would like to kindly push back on this assessment, and justify our choice of sampling parameters based on standards established in prior works, and sufficient evidence of robustness to the choice of sampling parameters. We also include new experiments with all remaining tasks using Qwen3-4B-instruct, to strengthen our argument. We hope these arguments together sufficiently address your concerns.
> > >
> > > *Pass@1 scores averaged over 4 seeds (except LCB) with developer-recommended sampling parameters. We use $K=4, N=16, T=10$ for RSA, and rejection sampling and majority voting are budget-matched per-prompt with RSA.*
> > > | Method | AIME-25 | HMMT-25 | LCB-v6 |  RG Games |  RG Cognition+ARC | SuperGPQA |
> > > |-----------|----------:|----------:|----------:| ----------:|----------:|----------:|
> > > | Reference |    47.71       |     30.63      |     49.16      |   57.12 | 45.26 | 42.67 |
> > > | Rejection Sampling    |    51.43       |     34.34     |  50.67  |  60.48      | 46.31 | 47.62 |
> > > | Majority voting    |     64.99      |     33.33      |     N/A   | 62.63    | 45.33 | **48.85** |
> > > | **RSA**   |    **68.49**       |    **44.43**       |    **55.66**       |  **68.88**  |  **56.84** | 48.13 |
> > >
> > > 1. **Developer-Recommended parameters are arbitrary**: The choice of temperature, among other sampling parameters, trades off between greedy decoding (temperature=0) and decoding from the untempered model (temperature=1). These are hyperparameters that are tuned based on tasks considered for model release (e.g. lower temperatures used for math, and higher temperatures for creative writing tasks). Thus, the sampling parameters provided are rough guidelines based on the evaluation criteria of the released models, not accounting for test-time scaling, and should not be taken as gold-standards. Since prior related literature provides ***no established guidelines for choosing these sampling parameters***, we believe this should not be the basis for rejection. In fact, it is often unclear how these parameters are chosen [4,5,6,8], while we rely on a reasonable default choice across tasks and models without any special tuning.
> > >
> > > 2. **Relative Improvement is consistent**: The key takeaway should be the relative improvement delivered by RSA across diverse tasks and model architectures, for a chosen set of default sampling parameters. The sampling parameters we choose are **not specially tuned**, but the **defaults with vLLM**. The breadth of experiments presented demonstrates that RSA consistently outperforms alternative methods, indicating that its performance is robust to variations in task-specific parameter choices.
> > >
> > > 3. **Ablation with recommended parameters**: To further alleviate the reviewer's concerns, we verified with Qwen3-4B-Instruct-2507's recommended sampling parameters, we still see RSA delivering consistent relative gains. While the exact scores differ slightly based on sampling parameters, the overall trends remain consistent with RSA, as expected since our original sampling parameters were not specially tuned. This, alongside our extensive experiments should solidify that RSA remains performant across a wide range of sampling parameters, and that our ***primary findings are consistent with the results produced when using the developer-recommended parameters.***
> > >
> > > 4. **Choice of our defaults**: Since the choice is *arbitrary*, and developer-recommended parameters are *different across models and tasks*, and further often *unknown for some models and tasks* (none have Reasoning Gym evaluation), we deliberately conducted all our experiments with the defaults of temperature=1.0 and top_p = 1 because it is a reasonable default which consistently works well across models, corresponding to the default pre-training distribution. Several prior works on test-time scaling use temperature=1 [4,7,9].

---

> > > > ### Author Response · Authors · 2025-11-25
> > > >
> > > > (continued from comment above)
> > > >
> > > > >  I am concerned about the fact that RSA is simply forcing non-thinking models to think more
> > > >
> > > > We appreciate the clarification, but do not agree that it is a weakness. In fact, forcing the model to think more is the primary feature of test-time scaling. The question then is *how* to best get the model to "think more". Recursive aggregation is one way to achieve this, which we demonstrate performs well compared to other verification-free strategies.
> > > >
> > > > **Discussion on [1,2,3]**: We already provide a comparison with self-verification [1] (Table 1: Rejection Sampling) and self-correction [2] (Table 2: Self-refinement). We have updated our draft to cite [2, 3] (Appendix B), while [1] was already cited in the original draft. Briefly, self-verification can be used as a parallel scaling method by generating multiple solutions in parallel and picking the best one by using the model itself as a verifer; thus these parallel scaling methods cannot think deeply and refine their responses. Self-correction is a sequential scaling method that only refines a single solution trajectory and can be prone to mode-collapse since it operates on refining, verifying and correcting only the current trajectory. RSA, on the other hand, is not explicitly templated to do any of these, it can generate a new solution by selecting reasoning fragments from **multiple** existing  solutions (Appendix F).
> > > >
> > > > We hope our responses, with the new experiments help address your concerns. Specifically regarding the new experiments, we argue that rerunning all models with new sampling parameters would incur substantial computational cost with little added value. The new Qwen3-4B-Instruct experiments show that the discussions in the paper would remain unchanged. Furthermore, it has not been standard practice to do this in the literature.
> > > >
> > > > [1]. Weng Y, Zhu M, Xia F, et al. Large language models are better reasoners with self-verification[C]//Findings of the Association for Computational Linguistics: EMNLP 2023. 2023: 2550-2575.
> > > >
> > > > [2]. Huang, Jie, et al. "Large language models cannot self-correct reasoning yet." arXiv preprint arXiv:2310.01798 (2023).
> > > >
> > > > [3]. Chen J, Ren J, Chen X, et al. Sets: Leveraging self-verification and self-correction for improved test-time scaling[J]. arXiv preprint arXiv:2501.19306, 2025.
> > > >
> > > > [4] Muennighoff et al. "s1: Simple test-time scaling." 	arXiv:2501.19393
> > > >
> > > > [5] Wang et al. "Mixture-of-Agents Enhances Large Language Model Capabilities." arXiv:2406.04692
> > > >
> > > > [6] Madaan et al. "Self-Refine: Iterative Refinement with Self-Feedback." arXiv:2303.17651
> > > >
> > > > [7] Agarwal, Sandhini, et al. "gpt-oss-120b & gpt-oss-20b model card." arXiv preprint arXiv:2508.10925 (2025).
> > > >
> > > > [8] Yang, An, et al. "Qwen3 technical report." arXiv preprint arXiv:2505.09388 (2025).
> > > >
> > > > [9] Huang, Audrey, et al. "Is Best-of-N the Best of Them? Coverage, Scaling, and Optimality in Inference-Time Alignment." Forty-second International Conference on Machine Learning.

---

### Official Review · Reviewer_YLr2 · 2025-10-31

**Soundness:** 2
**Presentation:** 2
**Contribution:** 1
**Rating:** 2
**Confidence:** 4

**Summary:**

This paper proposes a sequential-parallel test-time scaling method for LLMs. In each step, the proposed method first samples multiple response in parallel and then combine random subsets of these responses as LLM inputs for the next step.

**Strengths:**

* This paper connects sequential-parallel test-time scaling for LLMs with evolutionary algorithms.

**Weaknesses:**

* The primary weakness is the omission of highly relevant previous work on sequential-parallel scaling [1-3]. The paper must provide a detailed comparison outlining the technical differences and performance margins relative to these existing methods.
* Specifically, the basic underlying idea presented here has already been explored in prior work [1], making the technical contribution and insights appear highly incremental.
* In addition, the claimed improvements are achieved through the use of an expensive RL process, which renders the comparison with non-RL baseline methods inherently unfair and diminishes the practical viability of the proposed approach.


[1] Wang, Fei, et al. "DynScaling: Efficient Verifier-free Inference Scaling via Dynamic and Integrated Sampling." arXiv preprint arXiv:2506.16043 (2025).

[2] Snell, Charlie Victor, et al. "Scaling LLM test-time compute optimally can be more effective than scaling parameters for reasoning." The Thirteenth International Conference on Learning Representations. 2025.

[3] Wang, Junlin, et al. "Think deep, think fast: Investigating efficiency of verifier-free inference-time-scaling methods." arXiv preprint arXiv:2504.14047 (2025).

**Questions:**

See above.

---

> ### Author Response · Authors · 2025-11-17
>
> We thank the reviewer for their feedback. We believe the review is based on several critical misunderstandings of our core contributions and hope that our response clarifies these points.
>
> > In addition, the claimed improvements are achieved through the use of an expensive RL process, which renders the comparison with non-RL baseline methods inherently unfair and diminishes the practical viability of the proposed approach.
>
> **In fact RSA does *not* require RL.** RSA is a test-time scaling method, which works with a wide range of models **without any additional training or fine-tuning**. Most of our results (Table 1, Figures 4,5,6,7) apply RSA on the base model **without any additional training**.
>
> In §5.4, we study the best way to combine RL training with RSA as an optional additional step. We propose aggregation-aware RL to push the performance further, and to address the distribution shift with standard RL. In Figure 1 (left), we have clearly marked the results with and without RL as a direct comparison. Figure 8 in §5.4 is the only other reported result that uses any RL training. In addition, our proposed RL training does not diminish practical viability, since most of these models are already RL post-trained on diverse tasks before release, and our proposed recipe can be very easily implemented to augment this training.
>
> Taken together, the experiments demonstrate that RSA works with the base model without any additional training, same as all the test-time scaling baselines we consider.
>
> > The primary weakness is the omission of highly relevant previous work on sequential-parallel scaling [1-3]. The paper must provide a detailed comparison outlining the technical differences and performance margins relative to these existing methods.
> Specifically, the basic underlying idea presented here has already been explored in prior work [1], making the technical contribution and insights appear highly incremental.
>
> We would first like to clarify the key novelty and innovation of our work:
> (1) recursive aggregation procedure which conditions the model on multiple responses from the previous rounds.
> (2) Aggregation-aware RL training scheme to further improve test-time performance, while mitigating performance degradation due to distribution shift.
>
> We thank the reviewer for pointing us to [1] -- we were not aware of this **concurrent** work. The test-time scaling procedure does resemble RSA, as it using subsets of generated responses to seed the next response in an iterative loop. However, we note that our experiments span a much wider variety of models (including reasoning, non-reasoning, as well as SSM-style models). Moreover, our work includes a comprehensive study of the various parameters of the procedure to identify their importance. In addition to the test-time scaling procedure, we also propose a RL pipeline for aggregation to further improve performance.
>
> We would also like to note that this is a fairly active area of research with several concurrent  works which we tried to cover: see, for example, L72 in the paper. While we inevitably missed some related work and we thank the reviewer for pointing it out, we do not believe it to be a weakness of our paper that we have missed some work that appeared on arXiv in the last three months (as also noted in the [ICLR guidelines](https://iclr.cc/Conferences/2026/ReviewerGuide)).
>
> We also already cite [2] and our taxonomy of test-time scaling methods (§2) is based on it. Specifically, in [2] the authors study how test-time search can provide performance gains at the level of additional training compute. The study focused mainly on the setting where an external verifier (in the form of a process reward model or outcome reward model) is available. In contrast, RSA does not require access to any external verifier.
>
> Finally, [3] is an empirical study comparing various existing test-time scaling approaches. Our novel test-time scaling approach RSA is not part of the study. In fact, [3] identifies majority voting as the strongest approach (§4.1.3), and in our experiments we show that **RSA consistently outperforms majority voting** on free-form tasks (i.e., not multiple-choice questions).
>
> We will make these distinctions clearer and incorporate a discussion of [1,3] in our paper.
>
> **We hope that our response has clarified the positioning of our paper among concurrent work and clarified the role of additional RL training. We hope that the reviewer will reconsider their score in light of this. We are happy to address any further concerns.**

---

> > ### Author Response · Authors · 2025-11-27
> >
> > As we enter the final week of the discussion phase, we would like to gently remind the reviewer to engage with our rebuttal. We believe our response has tackled the core weaknesses raised in the review. Since this is the lowest score among the reviews in the batch, we would greatly appreciate it if the reviewer could acknowledge whether we have sufficiently addressed their concerns.

---

### Official Review · Reviewer_fcRM · 2025-11-01

**Soundness:** 3
**Presentation:** 3
**Contribution:** 3
**Rating:** 6
**Confidence:** 3

**Summary:**

The work proposes Recursive Self-Aggregation (RSA), a hybrid test-time scaling method that combines the benefits of both parallel and sequential scaling. Specifically, it recursively combines multiple reasoning chains and refines a population of candidates through repeated aggregation. This allows the models to reuse correct intermediate steps and self-correct the mistakes. RSA outperforms existing across diverse reasoning tasks such as math and code domains. The authors also propose integration with reinforcement learning by incorporating additional candidates in the optimization objective.

**Strengths:**

* The proposed RSA method is a novel and effective test-time scaling method that combines the benefits of both parallel and sequential scaling.
* Experiments are comprehensive: results on multiple benchmarks, such as HMMT, LiveCodeBench-v6, and SuperGPQA, demonstrate effective performance improvements over existing methods. The fact that applying RSA to a small model (Qwen3-Instruct-4B) can surpass a large model (DeepSeek-R1) is a significant finding.
* The writing is clear and easy to understand and follow.

**Weaknesses:**

* While RSA matches generation budgets with baselines, the paper does not provide detailed measurements of actual inference-time latency, which are crucial in real-world deployments. Note that methods such as majority voting should not increase the latency if conducted in parallel, while the proposed method might increase the latency, even if compute is not an issue.
* RSA's performance relies on carefully designed aggregation prompts. The paper lacks a principled or automated way to set aggregation prompts for new tasks or models.
* Lack of discussions and/or comparisons with other test-time scaling methods. For example, adaptive parallel scaling for reasoning methods, such as APR [1] and Multiverse [2], are also adaptive and could reason sequentially and in parallel. These methods demonstrate improved pareto frontier. Discussions and comparisons with these methods are important to highlight the advantages of RSA.

[1] Learning Adaptive Parallel Reasoning with Language Models. Pan, et al. https://arxiv.org/abs/2504.15466

[2] Multiverse: Your Language Models Secretly Decide How to Parallelize and Merge Generation. Yang, et al. https://arxiv.org/abs/2506.09991

**Questions:**

* Could the authors provide a detailed analysis of the latency of the proposed method, compared to the baseline methods? This would clarify RSA’s practicality under real deployment constraints and help quantify compute–performance trade-offs.
* Given the same base model, does the proposed method achieve an improved pareto frontier compared to adaptive parallel scaling methods, especially Multiverse, which solves Math reasoning problems as well?

---

> ### Author Response · Authors · 2025-11-17
>
> We thank the reviewer for their feedback and hope that our response addresses all the concerns raised.
>
> > While RSA matches generation budgets with baselines, the paper does not provide detailed measurements of actual inference-time latency, which are crucial in real-world deployments. Note that methods such as majority voting should not increase the latency if conducted in parallel, while the proposed method might increase the latency, even if compute is not an issue.
>
> While inference-time latency is an important consideration in some applications, we would like to emphasize that our work focuses on pushing the capabilities of models in settings where their performance is paramount even at the cost of added latency. For instance, consider competitions such as IMO or ICPC, where using additional compute time for better reasoning is preferable over lower latency with worse performance.
>
> Further, as discussed in §2, parallel scaling methods allow for optimal GPU memory utilization at the cost of the ability to think deeply. RSA is a hybrid scaling method that uses both parallel and sequential scaling. This is an inherent trade-off between reasoning performance and compute/latency.
>
> That said, with limited compute resources, even parallel scaling methods like majority voting are compute bounded. In particular, for AIME evaluation, in the budget-matched setting with 4 A100 GPUs, we find that RSA ($K=4, N=16, T=10$) is actually a little faster than majority voting -- RSA takes 6039 seconds, while majority voting takes 7171 seconds. We attribute this to the fact that aggregated responses are slightly shorter on average than reference proposals. However, vLLM batching could also play a role here.
>
> > RSA's performance relies on carefully designed aggregation prompts. The paper lacks a principled or automated way to set aggregation prompts for new tasks or models.
>
> We would like to clarify that the prompt used for RSA (Figure 3) was not specifically handcrafted and only contains some simple instructions; this is discussed in Appendix F. Moreover, **we used the same aggregation prompt (which was not refined or fine-tuned) across all our results spanning multiple tasks and models**, which demonstrates that RSA does not require specifically designed prompts for new tasks or models. While the performance can be possibly improved on certain tasks by handcrafting specific prompts, we chose to use a general prompt that simply instructs the model to aggregate multiple candidate solutions. We believe that combining RSA with prompt-optimization approaches like GEPA (https://arxiv.org/abs/2507.19457) could be an interesting avenue to explore in future work.
>
> > Lack of discussions and/or comparisons with other test-time scaling methods. For example, adaptive parallel scaling for reasoning methods, such as APR [1] and Multiverse [2], are also adaptive and could reason sequentially and in parallel. These methods demonstrate improved pareto frontier. Discussions and comparisons with these methods are important to highlight the advantages of RSA.
>
> Thank you for pointing out these related papers. We were unaware of them earlier but will include a discussion about them in the paper. However, a comparison with RSA would not be fair because RSA is a test-time scaling strategy applicable to off-the-shelf models, while both the listed methods require fine-tuning on curated datasets as described below.
>
> APR [1] proposes an approach to train models to perform sequential and parallel reasoning. The approach relies on first generating SFT data to train the model to use the `spawn` and `join` operations followed by reinforcement learning training. On the other hand, RSA can be used with any existing model **without any additional training**. Moreover, our RL training pipeline does not require any additional SFT or data curation. Additionally, APR is only tested on Countdown, and it is unkown how well it generalizes to other tasks, whereas RSA works across tasks from different domains (math, reasoning, code) without any training.
>
> Similarly, Multiverse [2] proposes attention mechanism for parallel thinking coupled with a data curation process for training models to utilize this mechanism. In contrast, RSA does not require any changes to the model architecture or inference engine and can work without access to curated fine-tuning data. The baselines in the paper are all test-time scaling strategies that are 1) external-verification-free and 2) applicable to off-the-shelf LLMs.
>
> **We hope that our response has addressed the reviewer's concerns about the latency of the approach, robustness to the prompt across models and relation to APR and Multiverse. We ask the reviewer to reconsider their score in light of our response, and we would be happy to address any further concerns.**

---

> > ### Author Response · Authors · 2025-11-27
> >
> > As we enter the final week of the discussion phase, we would like to gently remind the reviewer to engage with our rebuttal. We would be happy to address any further concerns the reviewer may have, please let us know if our comment has addressed the three weaknesses raised in the review.

---

### Official Review · Reviewer_YoB8 · 2025-11-02

**Soundness:** 3
**Presentation:** 3
**Contribution:** 3
**Rating:** 6
**Confidence:** 3

**Summary:**

This paper introduces a inference time paradigm called Recursive Self-Aggregation. In each step of RSA, N candidate solutions is kept and N subset of size K will be sampled and aggregated to get N new solutions for next step. Fixing the sampling budget, RSA achieves better performance than majority voting and self-refinement by a sizable margin. Furthermore, while standard long cot training hurts the performance after RSA, the authors propose an aggregation aware RL pipeline that can improve the end-to-end performance of RSA after training.

**Strengths:**

1. The paper clearly illustrated the proposed method.

2. The method is tested across multiple benchmarks and settings and the improvement over baseline is significant.

3. The proposed aggregation-aware RL show that this method can be used in training time as well.

**Weaknesses:**

1. As the authors noted, the method here bear certain similarity with the mixture-of-agents (MoA) paper. Either MoA or self-MoA should be included as a stronger baseline.

2. It is unclear why standard RL training will hurt the RSA performance. This may limit the usability of the methods in other settings.

**Questions:**

Please refer to the weakness for the major questions. Some other questions include

1. The largest K in the paper is only 4. Is the bounding factor here context length?

2. Given a small K, the information mixing between different responses may be slow. How should T and N be scaled jointly?

---

> ### Author Response · Authors · 2025-11-17
>
> We thank the reviewer for their comments and questions and do our best to address them all below.
>
> > As the authors noted, the method here bear certain similarity with the mixture-of-agents (MoA) paper. Either MoA or self-MoA should be included as a stronger baseline.
>
> Both MoA and RSA involve iterative aggregation of reasoning chains. We highlight the major differences between these algorithms:
> 1. Unlike RSA, MoA is **not** an evolutionary algorithm. RSA maintains a population of candidates, iteratively evolved through random recombination. MoA, on the other hand generates $K$ candidate improvements on each step conditioned on a single reasoning chain using the proposal ensemble, which are then aggregated to the next iteration's single reasoning chain using the aggregation model.
> 2. As a result of this single reasoning chain bottleneck, MoA relies on the inherent diversity of LLMs in the proposal ensemble, while RSA does not require this.
>
> We cannot directly compare MoA and RSA because the former requires a model ensemble. Instead, as the reviewer suggested, **we have added new experiments with self-MoA**. RSA performs significantly better as expected, but self-MoA is a strong baseline comparable with majority voting. We will include these results in the paper.
>
> *Pass@1 scores averaged over 4 seeds with Qwen3-4B-Instruct. We use K=4, N=16, T=10 for RSA, and K=4, T=10 for self-MoA.*
> | Method | AIME-25 | HMMT-25 |
> |-----------|----------:|----------:|
> | Reference |    43.91       |     27.17      |
> | Majority voting    |     68.33      |     35.0      |
> | Self-MoA    |     62.5      |     42.5      |
> | **RSA**   |    **73.18**       |    **47.55**       |
>
> >It is unclear why standard RL training will hurt the RSA performance. This may limit the usability of the methods in other settings.
>
> **Why standard RL hurts RSA:** As hypothesized in §4, standard RL induces a distribution shift between training with no aggregation and test-time RSA, where the policy is used as an aggregator. This is confirmed by our experiments, since simply augmenting the training dataset with aggregation prompts significantly improves performance (§5.4)
>
> **But this does *not* limit its usability:** Note that all models we evaluate with RSA in §5.1 and §5.2 are *already* RL post-trained to some extent (by the model developers), and as demonstrated in §5.4, even policies trained with standard RL greatly benefit from RSA. Furthermore, the greatest improvements come from RSA itself, not from additional RL training.
>
> It is not surprising that RL fine-tuning may need to be done differently depending on the inference strategy (for example, reasoning models are not fine-tuned in the same way as ones that will be used for direct prompting). Our study of Aggregation-aware RL is intended to answer the question: "What is the right way to fine-tune models knowing they will be used with RSA?" We find that through an extremely simple data-augmentation strategy during RL post-training, we can benefit from both RL and RSA gains.
>
> In fact, we argue that it is a shortcoming of many other test-time scaling papers that do not show how their performance is impacted by RL. For example, best-of-N test-time scaling performance can degrade after RL compared to the reference policy (https://arxiv.org/abs/2504.13837).
>
> > The largest K in the paper is only 4. Is the bounding factor here context length?
>
> Yes, the context length is the major constraint for aggregation. We note that this limitation is handled through increasing the sequential budget $T$ with RSA, each iteration increasing the effective aggreagtion size well beyond $K$. This is why going up from $K=1$ to $K=2$ produces the most significant gains.
>
> > Given a small K, the information mixing between different responses may be slow. How should T and N be scaled jointly?
>
> We expand upon the discussion at the end of §5.3 surrounding the results in Figures 5 and 6. Regardless of the values of $K$ and $N$, increasing $T$ consistently improves performance. The "optimal" relative scaling between $N$ and $T$ thus depends on the time constraints the user is limited by, and the specific task under consideration. The key observation is that setting $N$ too large for a fixed $T$ can hurt performance due to slow mixing, but increasing $T$ for a fixed $N$ will not. Being specific, assuming we set $T=10$ as in the paper, our experiments show that we roughly want to scale $N = 4K$.
>
> **We would be happy to further engage with the reviewer to address any other concerns. We kindly request the reviewer to modify their scores accordingly if they are satisfied with our response.**

---

> > ### Author Response · Authors · 2025-11-27
> >
> > As we enter the final week of the discussion phase, we would like to gently remind the reviewer to engage with our rebuttal. We have included self-MoE experiments and tried to address the other concerns expressed by the reviewer.

---

> > > ### Comment · Reviewer_YoB8 · 2025-11-27
> > >
> > > I would like to thank the authors for the detailed rebuttal. Most of my concerns are addressed and I would raise the score to 8.

---

### Author Response · Authors · 2025-12-04

Dear AC,

Thank you for your additional effort in reviewing our submission given the current situation at the conference. To assist in your decision-making, we have provided a summary below of the concerns raised by the reviewers and how we addressed them during the rebuttal process.

**Reviewer YoB8:** The reviewer praised the paper's clarity and the significant performance improvements from RSA. They raised concerns about the lack of a MoA-style baseline and requested clarification on the role of our proposed RL training recipe and the parameters $N$, $T$, and $K$.

In our rebuttal, we provided a concrete comparison between MoA and RSA, including new empirical results showing that RSA consistently outperforms MoA. We also provided clarifications on the role of our RL training scheme and the various parameters based on ablations in the paper. **We note that the reviewer increased their score to 8 following our response.**

**Reviewer fcRM:** The reviewer noted the novelty and effectiveness of the proposed approach, along with the comprehensive experiments. They raised concerns regarding inference-time latency, the choice of the aggregation prompt, and a couple of missing related works.

In our rebuttal, we clarified that while the focus of the paper is on problems requiring *longer thinking*, the **practical latency of RSA is still lower than compute-matched baselines like majority voting**. We also clarified that the prompt is minimal with basic instructions and was not specifically tuned. Finally, we provided a detailed comparison to the related work suggested by the reviewer, noting that those works mainly rely on specific datasets and architectural modifications, whereas RSA works with any given language model without modification or training.

**Reviewer YLr2:** The reviewer mainly focused on similarities to **concurrent** related work and misunderstood the role of RL training for RSA.

We described the concrete differences between our work and the mentioned related work, including experiments on more tasks with more base models. We also clarified that RSA works effectively without any additional RL training and that most of our results use base models; the proposed aggregation-aware RL training only further improves performance. Finally, we provided a detailed comparison to the other related work mentioned by the reviewer (some of which was already included in the paper).

**Reviewer Jzh2:** The reviewer noted the comprehensive experiments and detailed ablation study but raised concerns about sampling parameters, the benefit to reasoning models, and compute-matching for baselines.

In our response, we noted that there is no single optimal choice of sampling parameters, especially when combined with test-time scaling methods (citing prior work that uses temperature 1, which we also use). Additionally, we provided empirical results with the developer-recommended parameters for Qwen3-4B on all tasks, showing that the relative performance gains from RSA are consistent across different sampling parameters. Further, we clarified that the goal of methods like RSA is precisely to increase thinking time and that compute-matching was performed on a per-prompt basis. **We note that the reviewer expressed their intention to increase their score if provided with additional results, which we included in our second response.**

We strongly believe that we have addressed every concern raised by the reviewers, backed by several new empirical results. We ask you to consider our responses when making your decision.

---

### Meta-Review · Area_Chair_5mc8 · 2026-01-09

**Summary:**

This work proposes Recursive Self-Aggregation (RSA), a hybrid test-time scaling method that combines the benefits of both parallel and sequential scaling. The reviewers raised the following concerns:

1. The distinctions between this work and prior approaches need to be more clearly articulated and better framed.

2. The experimental results are not sufficiently strong to convincingly demonstrate the advantages of the proposed method over existing alternative solutions.

Although the proposed approach is promising, the paper is not recommended for acceptance in its current form. The authors are encouraged to address the reviewers’ feedback and further strengthen the work for resubmission to other venues.

**Reviewer Scores:**

NA

---

### Decision · Program_Chairs · 2026-01-26

Reject